



# Vectorized dataset of check dams on the Chinese Loess Plateau using object-based classification method from Google Earth images

Yi Zeng[1, 2][¶], Tongge Jing[1, 2][¶], Baodong Xu[3], Xiankun Yang[4], Jinshi Jian[1, 2], Renjie Zong[1, 2], Bing Wang[1, 2], Wei Dai[1, 2], Lei Deng[1, 2], Nufang Fang[1, 2*], Zhihua Shi[3]

[1]State Key Laboratory of Soil Erosion and Dryland Farming on the Loess Plateau, Institute of Soil and Water Conservation, Northwest A&F University, 26 Xinong Road, Yangling, Shaanxi Province 712100, PR China
[2]Institute of Soil and Water Conservation, Chinese Academy of Sciences and Ministry of Water Resources, 26 Xinong Road, Yangling, Shaanxi Province 712100, PR China
[3]College of Resources and Environment, Huazhong Agricultural University, Wuhan 430070, PR China
[4]School of Geographical Sciences, Guangzhou University, Guangzhou, 510006, PR China

[¶]These authors contribute equally

*Correspondence to*: Nufang Fang (fnf@ms.iswc.ac.cn)

**Abstract.** The Chinese government has invested tens of billions of dollars and about 60 years to implement a large-scale
check dam project on the Chinese Loess Plateau (CLP) to control severe soil erosion. These check dams have trapped billions of tons of eroded sediment over the past few decades, significantly reducing the sediment load of the Yellow River, which was once the river with the largest sediment load in the world. However, there is still great uncertainty about how much sediment is trapped by check dams and what roles they play in the flow and sediment variability in the Yellow River, because the number and spatial distribution of check dams are still unclear. In this study, we produced the first vectorized
dataset of check dam on the CLP, combining high-resolution and easily accessible Google Earth images with object-based classification methods. We first investigated and analysed the key characteristics of check dams, and obtained the 0.3-1.0 m resolution Google Earth image of the best extraction period. Then we preliminarily obtained the rough check dam layer through multi-scale segmentation, threshold classification, and river network superposition. Finally, a self-developed human-computer interaction program combined with auxiliary data, visual interpretation, and expert knowledge is used to improve
the classification accuracy of check dams. The accuracy of the dataset is verified by 1947 collected test samples, and the producer's accuracy and user's accuracy of the check dam are 88.9% and 99.5%, respectively. Furthermore, at the provincial level, the area and number of check dams in our dataset are highly consistent with the latest official statistics of check dams, with $R^2 > 0.99$. Our study provides fundamental dataset for accurately assessing the ecosystem service functions of check dams, including sediment retention, carbon sequestration, grain supply, and will help to interpret current changes in sediment
delivery of the Yellow River and plan future soil and water conservation projects. The check dam dataset introduced in this article is freely available at https://doi.org/10.5281/zenodo.7857443 (Zeng et al., 2023).



# 1 Introduction

Accelerated soil erosion has caused worldwide land degradation, water pollution, and grain yield reduction, which has become one of the most pressing environmental issues threatening human sustainable development (Alewell et al., 2020; Wuepper et al., 2020). To alleviate serious soil erosion, many countries have implemented a series of soil and water conservation projects according to local conditions in recent years, including vegetation restoration, conservation tillage (Klik and Rosner, 2020), terrace and check dam construction (Klik and Rosner, 2020; Liu et al., 2021b; Osman and Sauerborn, 2001). Among them, check dams have played an amazing role in soil and water conservation and are widely applied in global arid and semi-arid regions with serious soil erosion, such as China, Spain, Australia, America, India, Iran, and Ethiopia (Abbasi et al., 2019; Lucas-Borja et al., 2021). The check dams built in the gullies can directly intercept the eroded sediment from the watershed (Zeng et al., 2023). Additionally, the silted land behind the check dam formed by the eroded sediment reduces the gully slope and runoff velocity, stabilizes the gully bed, and weakens erosion kinetic energy (Wang et al., 2021). The "sediment retaining" and "erosion reduction" of the check dam synergistically reduce the watershed sediment yield. According to data from different study areas such as China, Spain, and America, check dams can reduce sediment yield by about 50-77% at the watershed scale (Boix-Fayos et al., 2008; Polyakov et al., 2014; Zhao et al., 2017). In addition to preventing soil erosion, check dams also provide more unexpected ecosystem services, including carbon sequestration and grain supply. Large amounts of organic-rich eroded sediments are buried and effectively preserved by check dams, becoming an important terrestrial carbon sink (Yao et al., 2022). Moreover, the fertile silted land behind the check dam, with its high water content and strong drought resistance, is often used as cropland with high and stable grain production (Zeng et al., 2022b).

The Chinese Loess Plateau (CLP) is the region with the largest number and the densest distribution of check dams in the world due to serious soil erosion in the past few decades (Zeng et al., 2022a). Since the 1970s, the Chinese government and local farmers have widely built check dams in the gullies to prevent sediment loss to the Yellow River, which used to have the largest riverine sediment flux in the world. Tens of thousands of check dams cooperate with other soil and water conservation measures such as vegetation restoration and terraces, have greatly reduced the sediment load into the Yellow River from the CLP by ~85% in recent decades (from ~1.6 Pg yr$^{-1}$ in the period 1919-1960 to ~0.25 Pg yr$^{-1}$ in the period 2010-2016) (Wang et al., 2016). However, there is still great uncertainty about how much sediment is intercepted by check dams and how many roles they play in the flow and sediment variation in the Yellow River because the number and spatial distribution of check dams are still unclear (Liu et al., 2021a). The Ministry of Water Resource of P.R. China reported that more than 110,000 check dams had been constructed and captured about $21\times10^9$ m$^3$ sediment on the CLP (CMWR, 2003). Yet, the Bulletin of First National Census for Water reported only 58,446 check dams had been constructed and remained intact till 2013 and the silt sediment is about $7\times10^9$ m$^3$ (CMWR, 2013). Almost all regional studies regarding the sediment budget of the Yellow River referred to one of these two reports, and the huge discrepancy between the check dam data in these reports results in that all sediment retention and carbon storage estimated using these data may be overestimated or



underestimated (Ran et al., 2018; Wang et al., 2016; Wang et al., 2011). Additionally, although large check dams designed with capacities between $5×10^5$ to $5×10^6$ m³ and funded by the central government had detail records, tens of thousands check dams constructed by local governments or farmers on the CLP remains unclear (Fang et al., 2019). Therefore, it is necessary to generate the spatial distribution data of check dams on the CLP, which is an important prerequisite to accurately quantify the contribution of check dams on the CLP to the variation of sediment load in the Yellow River and to evaluate the carbon

sequestration and grain supply benefits of check dams.

The early check dam data mainly come from the field survey organized by government departments, which may lead to incalculable statistical errors at the regional scale (Jin et al., 2012). In recent decades, remote sensing has been widely used to obtain regional, national, and even global land cover or land use, and has made great progress in cropland, terraces, water identification, and so on (Cao et al., 2021; Pazúr et al., 2022; Tortini et al., 2020). However, it is incredible that, although the

dataset of check dams is very important, only two studies have explored the possibility of obtaining check dam data based on remote sensing technology (Li et al., 2021; Tian et al., 2013). Zhao et al. (2013) obtained the spatial distribution of check dams in the Huangfuchuan watershed based on the supervised classification and Landsat images. Li et al. (2021) proposed a method integrating deep learning and object-based classification to extract check dams within a range of tens to hundreds of km². Nevertheless, these studies have only explored different methods to extract check dams on a very small scale, and could

not be extended to the whole CLP. That is, the current dataset of check dams on the CLP is still blank.

Several key difficulties limit the current extraction of check dams at the regional scale. Firstly, check dams on the CLP are mainly small and medium-sized check dams, with a corresponding silted area of 0.2-2 hm² (Bai et al., 2020). The decametric-resolution images (e.g., Landsat-7/8) may produce large silted land extraction errors due to the limited number of pixels and jagged edges (Ling et al., 2019). Secondly, the silted land formed by check dams is usually used for planting

crops (Li et al., 2019a), which is difficult to distinguish from the surrounding slope cropland and terrace in terms of spectral characteristics. Finally, silted land has the characteristics of large spatial heterogeneity and patch fragmentation, which is more difficult to identify than other land use types (Li et al., 2021). The traditional pixel-based extraction methods usually focus on medium resolution images, and seldom consider the structure and texture within the category and the correlation information between adjacent pixels (Duro et al., 2012; Hussain et al., 2013). Meanwhile, this method also produces salt-

and-pepper noise, which reduces the integrity of classification (Zhang et al., 2022). In contrast, the object-based classification method comprehensively considers a series of factors, such as spectral statistical features, shape, size, texture, and adjacency, and can obtain high-precision silted land extraction results combined with high-resolution images (Belgiu and Csillik, 2018). Therefore, combined with high-resolution and easily accessible Google Earth images and object-based classification, we provide the check dam dataset on the CLP for the first time. The self-developed computer program

combined with auxiliary data, visual interpretation and expert knowledge is used to improve the extraction accuracy of check dams. The accuracy of this dataset is verified by a visually interpreted test set and the latest official statistics. This dataset provides basic data for researchers to quantify the ecosystem service function of check dams, and to provide useful information for policy makers to plan soil and water conservation projects.





## 2 Method

It should be emphasized that the extraction of check dams on the CLP is not to extract the dam body, but to extract the silted land behind the check dam formed by the eroded sediment (hereinafter referred to as dam land). Because the sediment retention, carbon sequestration, and grain supply of check dams can only be determined if the dam land area is obtained. According to previous reports, the hilly and gully region occupies 33% (21.2 km$^2$) of the area of the CLP (64.0 km$^2$), but about 85% of the check dams are distributed here (Liu et al., 2021a). Therefore, to improve efficiency, we divide the CLP

into dam concentrated region (hilly and gully region) and dam sparse region (other region) according to the spatial distribution characteristics of check dams. For the dam concentrated region, we extracted dam land mainly through high-resolution Google Earth images and object-based classification method (Fig. 1). However, the area of dam sparse region is too large, and the acquisition and processing of corresponding Google Earth images are complex. In addition, only about 15% of the check dams are distributed in this region. Therefore, we extracted the dam land by artificial visual interpretation in

Google Earth. Noticeable, there are no check dams distributed in the Mu Us Desert in the northwest, the Guanzhong Plain in the middle, and the Rocky Mountains in the east of the dam sparse region (Fig. 2a) (Jin et al., 2012), so we masked these areas, which greatly reduced the workload of visual interpretation. Finally, we aggregate the dam land extraction results of these two regions and verify the accuracy.

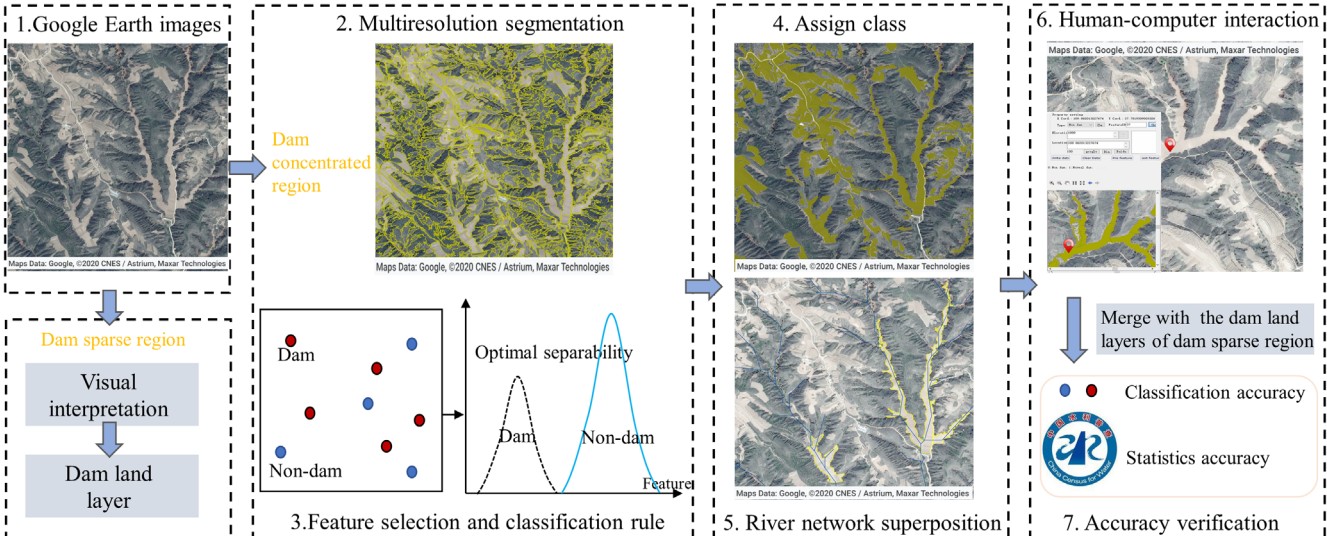

**Figure 1: The workflow of check dam land extraction on the Chinese Loess Plateau.**

## 2.1 Satellite images collection

Dam land is one of the most productive land types on the CLP, so the vast majority of dam land on the CLP is cultivated. Therefore, the extraction object of this study is the cultivated dam land on the CLP, excluding water-covered dams and abandoned dam land. Farmers on the CLP usually plough the dam land around May (Zhang et al., 2019), while the gullies



around the dam land are covered with vegetation, which makes it easy to distinguish the dam land from the surrounding landscape from satellite images. A large number of field investigations and Google Earth image observations have also confirmed this phenomenon (Fig. 2). Additionally, we randomly selected dam land ($N_{dam}$=127) and surrounding landscape samples ($N_{dam}$=130) in three counties with dense distribution of check dams (Baota county: $N_{dam}$=41 and $N_{sur}$=48; Zizhou county: $N_{dam}$=37 and $N_{sur}$=45; Lin county: $N_{dam}$=49 and $N_{sur}$=37). Then, combined with the Sentinel-2 image, the NDVI time

series of these sample points in 2019 are calculated in Google Earth Engine. We also found that there was the most significant difference in NDVI between the dam land and the surrounding landscape around May (Fig. 3). Therefore, we collected all available May images from 2016 to 2020 in the dam concentrated region in Google Earth, with a spatial resolution of 0.3-1.0 m. It is worth noting that the images in Google Earth are usually spliced from images taken in different periods or from different satellites, resulting in the possible chromatic aberration between the images of the two adjacent

scenes (Li et al., 2019b). To avoid subsequent segmentation errors caused by image chromatic aberration, we divided the images of the dam concentrated region according to the shooting date and obtained a total of 52 images.

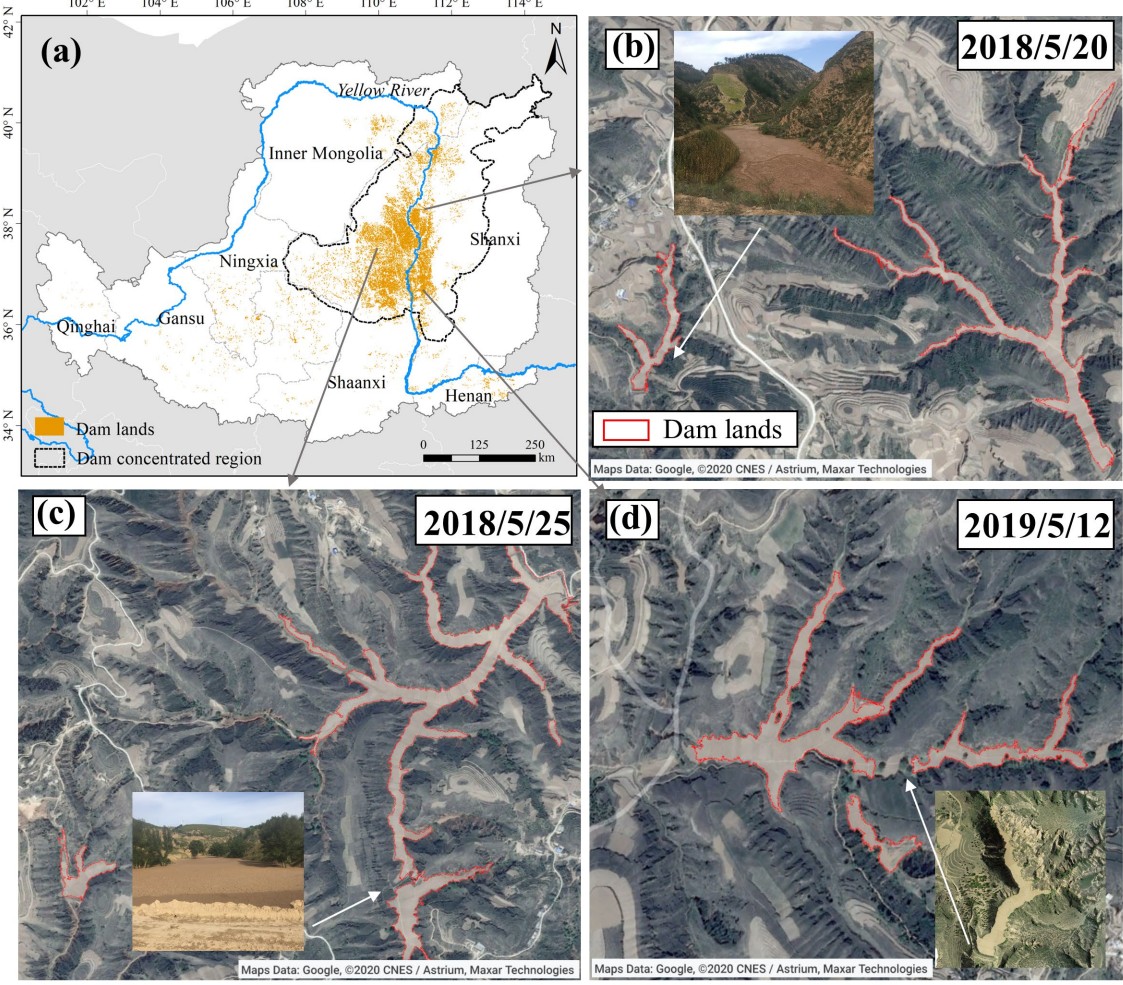

**Figure 2: Check dams in the study area. (a) Dam concentration region on the Chinese Loess Plateau, (b-d) Google Earth images, photographic images, and unmanned aerial vehicle images of dam lands in May.**

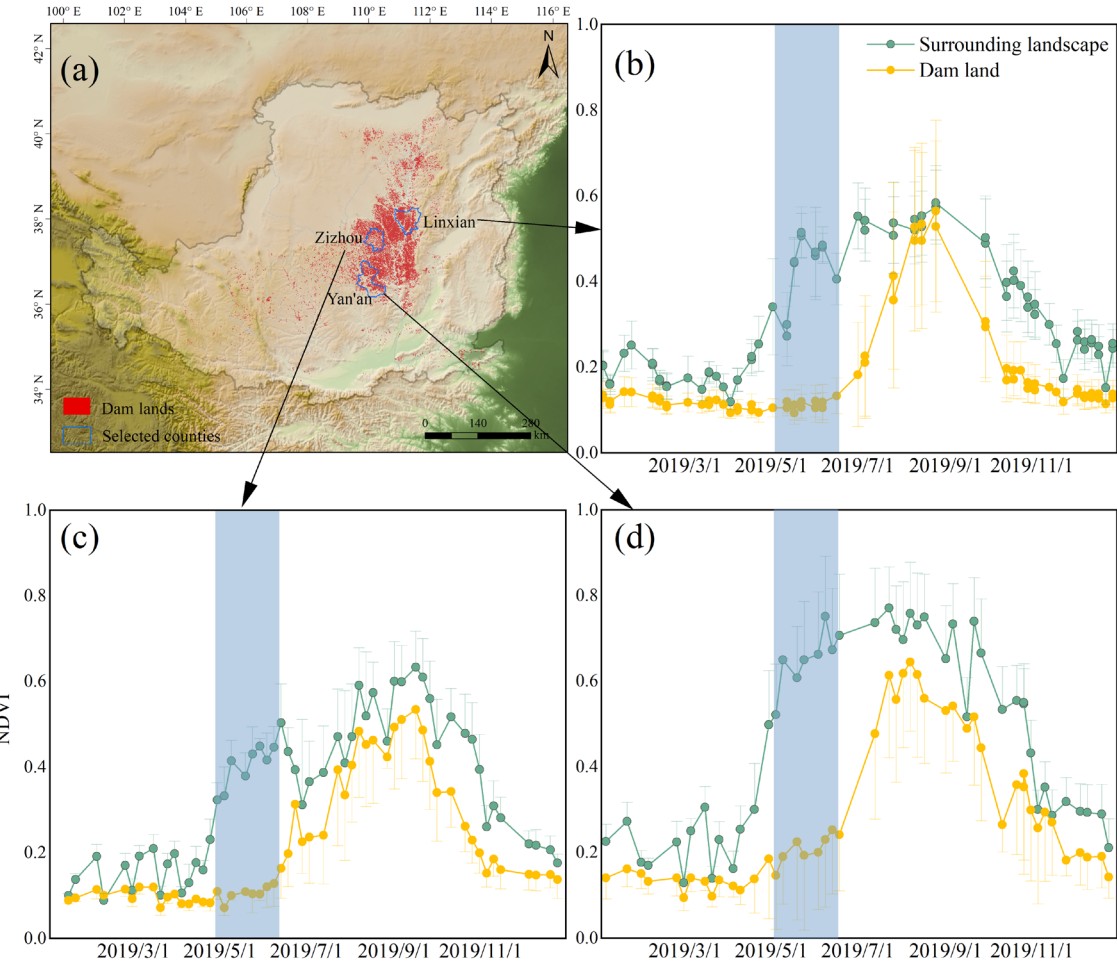


**Figure 3: (a) Location of the three selected counties, (b-d) NDVI time series of the dam land and surrounding landscape of the three selected counties. The blue rectangle represents the period with the largest NDVI difference between the dam land and the surrounding landscape.**

## 2.2 Object-Based Classification Method

### 2.2.1 Multiresolution segmentation

Here we used multiresolution segmentation algorithm in the eCognition Developer. This segmentation method achieves image segmentation by merging adjacent pixels with similar features under the premise of ensuring the maximum homogeneity between the pixels within the object (Gupta and Bhadauria, 2014). The scale, shape, and compactness parameters are the most important parameters in multiresolution segmentation algorithms, and their values will affect the

segmentation results (Munyati, 2018). The scale parameter is used to determine the maximum heterogeneity of the generated

object and to control the size of the segmented object. The estimation of scale parameter 2 (ESP2) plugin in eCognition Developer can automatically evaluate the segmentation effect based on the local variance (LV) and its rate of change (ROC). Therefore, we used the ESP2 plugin to determine the optimal scale parameter for dam land extraction (Wei et al., 2021). The setting of the shape and compactness parameters is also crucial for the segmentation and subsequent classification of dam

land, as the dam land is usually narrow and irregular in shape. Therefore, we made different combinations of these parameters to test the best parameter settings in combination with visual inspection. Finally, we set the scale parameter, shape weight, and compactness weight to 100, 0.7, and 0.3 to obtain the best segmentation results (Fig. 4).

**Figure 4: Multiresolution segmentation of dam land. (a) Original Google Earth image; (b) Object mean view in eCognition**
**Developer; (c) View after multiresolution segmentation; (d) Classify the segmented objects. The yellow polygon is the extracted target category.**



### 2.2.2 Feature selection and classification rule

Here we used the assign class algorithm in the eCognition Developer, which determines the category of segmented objects by setting threshold conditions. When there were significant differences between the target and background categories for some features, the assign class algorithm could be used to construct classification rules (Tamta et al., 2015). The assign class algorithm was more suitable for our study than other classification methods, as our goal was to include more dam land in the target category. For each segmented image, we first randomly selected 30-80 dam land and non-dam land at different locations according to the size of the image. Then, we used the separability and thresholds algorithm in eCognition Developer to automatically select classification features (e.g., red, green, and blue band, shape, texture, and brightness) and determine the threshold of the selected feature (Guirado et al., 2019) (Fig. 1). Finally, we used the feature threshold to classify the segmented objects and manually adjusted the threshold range combined with visual interpretation to ensure that all dam land was included in the classification range. Through the above steps, we obtained the bare land layer of the dam concentrated region in May, mainly including the dam land in gullies and the cropland on slope land (Fig. 4).

### 2.3 River network superposition and human-computer interaction

All dam lands on the CLP are distributed in gullies, which can be easily separated from non-dam land on the slope through the river network. We first used SRTM-DEM with a resolution of 30 m to extract river networks in ArcGIS. Then we superimpose the river network and the bare land layer to extract the dam land in the gullies (Fig. 5a, b). To further improve the accuracy of dam land extraction, we developed a convenient human-computer interaction program to mark and eliminate incorrect classifications. The left window of the program can call the high-resolution Google Earth image, and the right side can call the dam land layer we extracted in the previous step (Fig. 5c). We assign values to each extracted vector polygon (e.g., dam land is 1, non-dam land is 2) based on auxiliary data, visual interpretation, and expert knowledge. Finally, we merge the vector polygons with a value of 1 obtained from 52 images in ArcGIS, which is the final dam land layer in the dam concentration region.



**Figure 5: (a) Classified objects superimposed on river network, (b) removal of non-dam land by superimposed river network; (c) human-computer interaction.**

**2.4 Accuracy verification**

At present, there is no available spatial distribution dataset of check dams on the CLP. Therefore, we take the test samples obtained from Google Earth by visual interpretation to verify our dataset. Due to the spatial distribution of check dams is not

uniform, the traditional uniform sampling may lead to the deviation of accuracy evaluation. To improve the reliability of verification, we determine the number of test samples in each county according to the number of check dams in the dataset at the county level. That is, we allocated more test samples in counties where check dams are concentrated. A total of 1947 test



samples were acquired within the study area, of which 949 samples were interpreted as silted dam land and 998 samples as non-dam land. We evaluated the accuracy of the check dam dataset by calculating the confusion matrix, including producer's
(PA), user's accuracy (UA), and Kappa coefficient. Additionally, the Ministry of Water Resources of the People's Republic of China has provided the area and number of check dams in different provinces on the CLP according to statistical data of county-level water conservancy departments (CMWR, 2013). We compare our check dam dataset with this official statistical data for mutual verification.

**2.5 Estimation of sediment volume of check dam**

Our recent research has established a method for estimating the sediment volume of check dams based on unmanned aerial Vehicle (UAV) photogrammetry and proposed an area-volume empirical formula for estimating the volume of check dams at regional scale (Zeng et al., 2022a). Based on this area-volume empirical formula, we can accurately estimate the sediment volume of check dams according to the area of each dam land. A more detailed description and accuracy verification of this sediment volume estimation method for check dam can be found in Zeng et al. (2022).

$$
V = \begin{cases}
(6.31 \pm 0.03) A_{area}^{(1.33 \pm 0.01)} & (0 \le A_{area} < 2) \\
(3.80 \pm 0.07) A_{area}^{(1.51 \pm 0.01)} & (2 \le A_{area} < 5) \\
(8.48 \pm 0.89) A_{area}^{(0.96 \pm 0.02)} & (5 \le A_{area})
\end{cases}
\tag{1}
$$

where V is the sediment volume of check dam ($\times 10^4$ m$^3$); Area is the silted land area (hm$^2$).

**3 Results and discussion**

**3.1 Accuracy assessment**

We first evaluated our check dam dataset with 1947 collected test samples. The results of the confusion matrix show that the
overall accuracy (OA) of the check dam map is 94.4%, in which the PA and UA of the check dam map are 88.9% and 99.5%, respectively. The non-dam land class had both UA and PA over 90%. Additionally, the kappa coefficient is 0.89, indicating that the dataset has high classification accuracy (Table 1).

**Table 1. Accuracy evaluation of check dam maps with 2000 random test samples.**

| Class | Non-dam land | Dam land | PA (%) | UA (%) |
|---|---|---|---|---|
| Non-dam land | 994 | 105 | 99.6 | 90.4 |
| Dam land | 4 | 844 | 88.9 | 99.5 |
| | OA=94.4% | | Kappa=0.89 | |



We further combined with official statistics of check dams to verify the overall accuracy of our dataset. At the provincial level, the dam land areas of our dataset are quite consistent with that in the Bulletin of First National Census for Water (CMWR, 2013), and the $R^2$ is 0.999 (Fig. 6a). Similarly, the number of check dams in our dataset shows very high correlation with the official statistics, with a slope of 1.1185 and $R^2$ of 0.995 (Fig. 6b). There results show that compared with the official statistics of check dams in 2003 (110,000 check dams), the official statistics of check dams in 2013 is more

reliable (58,446 check dams). Additionally, the high reliability of our check dam dataset is also confirmed through mutual verification.

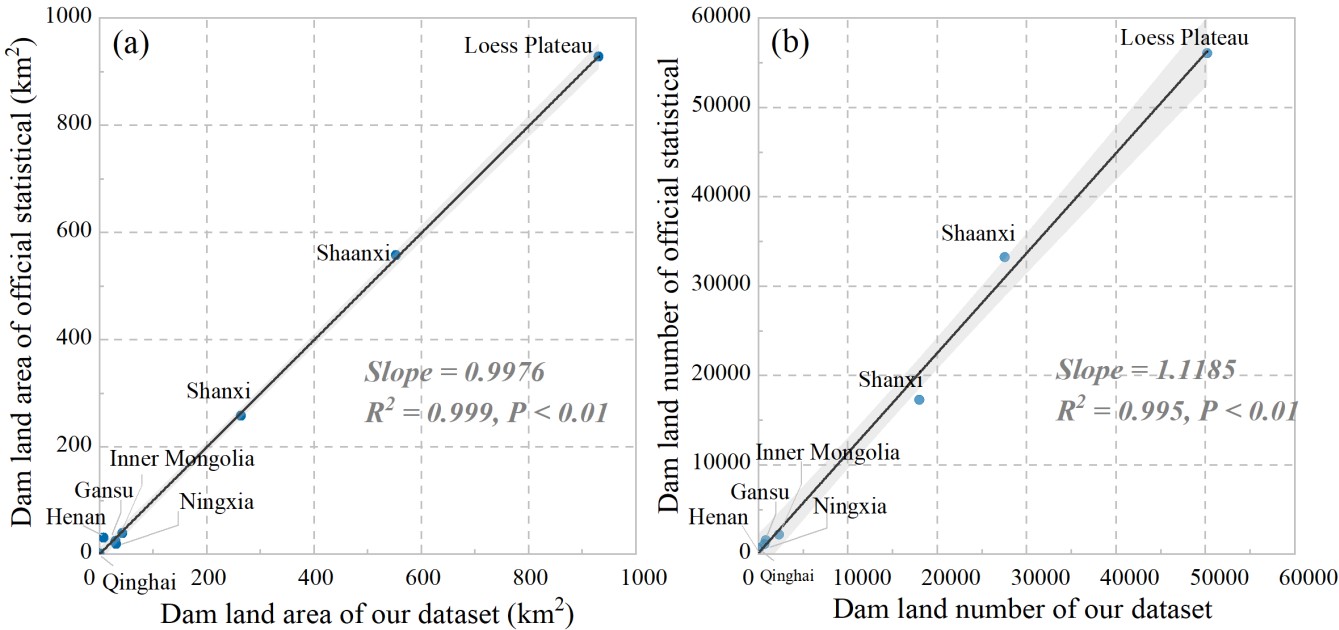

**Figure 6: Comparison with the area (a) and number (b) of official statistics of check dams. The grey shade areas represent 95% confidence interval.**


## 3.2 Spatial distribution, silted area, and sediment volume of check dams

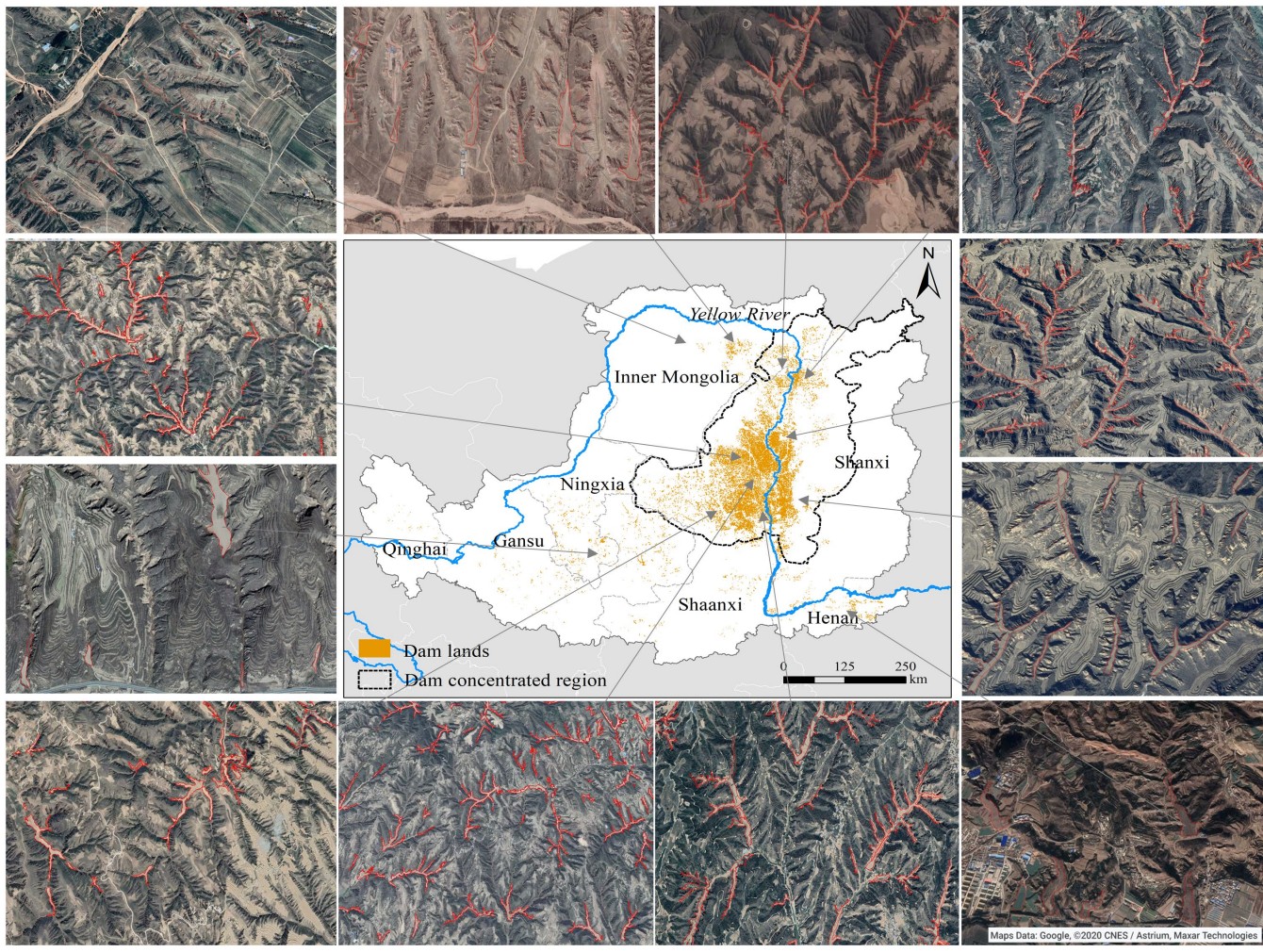

**Figure 7: Spatial distribution of check dams on the Chinese Loess Plateau.**

A total of 50,226 existing check dams were identified. The number of check dams in Shaanxi, Shanxi, Inner Mongolia, Henan, Gansu, Ningxia, and Qinghai are 27,557, 18,000, 2,288, 833, 827, 653, and 68, accounting for 54.9%, 35.8%, 4.6%, 1.7%, 1.6%, 1.3%, and 0.1% of the number of check dams on the CLP, respectively (Table 2). Check dams are mainly distributed in Northern Shaanxi and Western Shanxi (Fig. 7), that is, the loess hilly and gully region, which is one of the most seriously eroded areas on the CLP and even in the world. The spatial distribution of check dams is closely related to regional soil erosion, which is in line with the soil conservation function of check dams on the CLP. According to the dam land area in the classification standard of check dams in the Regulation of techniques for comprehensive control of soil erosion-Technique for erosion control of gullies (GBT16453.3-1996), the numbers of the micro (dam land area ranges from 0-0.2 hm$^2$), small (0.2-2 hm$^2$), medium (2-7 hm$^2$), and large check dams (>7 hm$^2$) are 4,207, 34,577, 9,192 and 2,250



respectively (Fig. 8). The small and medium check dams with the highest proportion account for about 69% and 18% of the total number of check dams, which is basically consistent with the results previously reported (70% and 18%) (CMWR, 2013).

**Table 2. Number, silted area, and sediment volume of check dams in different regions on the Chinese Loess Plateau.**

| Region | Number | Area (km²) | Volume (×10⁶ m³) |
|---|---|---|---|
| Shaanxi | 27557 (54.9%) | 553.6 (59.5%) | 3946.5 (60.0%) |
| Shanxi | 18000 (35.8%) | 264.0 (28.4%) | 1814.3 (27.6%) |
| Inner Mongolia | 2288 (4.6%) | 42.7 (4.6%) | 308.1 (4.7%) |
| Henan | 833 (1.7%) | 8.5 (0.9%) | 55.4 (0.8%) |
| Gansu | 827 (1.6%) | 30.2 (3.2%) | 212.8 (3.2%) |
| Ningxia | 653 (1.3%) | 31.7 (3.4%) | 234.3 (3.6%) |
| Qinghai | 68 (0.1%) | 0.5 (0.1%) | 3.3 (0.1%) |
| Chinese Loess Plateau | 50226 | 931.0 | 6574.7 |

The dam land area of all check dams on the CLP is 931 km², of which Shaanxi province (553.6 km²) and Shanxi province (264.0 km²) account for about 88% (Table 2). The existing 50,226 check dams intercept a total of 6.6 billion m³ of sediment (Table 2). If the bulk density of dam land is estimated as 1.5 g cm⁻³ previously reported (Fang et al., 2019), check dams intercept about 10.2 billion tons of sediment, which equals about 46% of the Yellow River's sediment load to the Bohai Sea during the period 1970-2020.

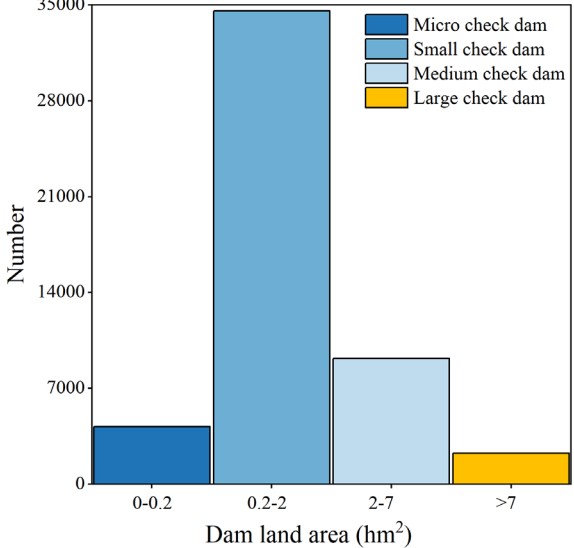

**Figure 8: Frequency distribution of dam land area of check dams. The classification of check dams adopts the classification standard of check dams in the Regulation of techniques for comprehensive control of soil erosion-Technique for erosion control of gullies (GBT16453.3-1996).**



### 3.3 Broader implications

In recent years, check dams on the CLP have been paid more and more attention by researchers because of their important role in soil and water conservation. Using "check dam" and "Loess Plateau" as keywords, 8450 research articles were
searched on Google Scholar, of which 3540 were published after 2018. Most studies to date have relied on time-consuming and laborious field investigations to find check dams suitable for the research objectives (Liu et al., 2022; Zhang et al., 2020). Our dataset can provide the precise spatial location of check dams, which greatly saves the manpower and material resources of researchers. More importantly, our dataset provides key parameters such as silted land area and sediment volume of check dams, which are crucial for those studies using check dams to calculate soil erosion rate (Wang et al., 2020), estimate
watershed sediment delivery ratio (Zeng et al., 2023), and validate the soil erosion model (Zhao et al., 2017).

For policymakers, our database will fill the gap in the current check dam dataset and provide unprecedented detailed data on the spatial distribution, silted land area, and sediment volume of check dams on the CLP. When these key parameters of check dams are obtained, we can conveniently and accurately evaluate the ecosystem services functions of check dams, including sediment retention (Zhang and She, 2021), carbon sequestration (Ran et al., 2014), and grain supply (Shi et al.,
2020). We preliminarily estimated that the 50,226 active check dams on the CLP have intercepted about 10.2 billion tons eroded sediment during the period 1970-2020, which equals 46% of the sediment load of the Yellow River, once the largest sediment contributor to the global ocean. These data will be beneficial to understand the anthropogenic influences on the unprecedented changes of sediment reduction in the Yellow River (Wang et al., 2016). Additionally, recent review studies emphasize the important carbon sequestration potential of check dams, and point out that there are still great uncertainties
and challenges in the estimation of carbon storage of check dams on regional scale (Yao et al., 2022). When the sediment carbon content of check dams is obtained by combining large scale filed sampling or literature compilation, the carbon storage of check dams can be easily estimated by using our check dam dataset (Fang et al., 2023). Moreover, the check dams silt 93,150 hm$^2$ of farmland has better moisture conditions than terraced and slope cropland. Previous study shows the grain yield of silted dam land is 2~3 times that of terraced fields and 6~10 times that of slope cropland (Shi et al., 2020; Zeng et al.,
2022b). Our check dam dataset can be used to determine the grain supply of check dams, combined with the grain yield per unit area of the silted dam land. Finally, according to the Chinese government's plan, 56,161 new check dams will be built on the CLP by 2030 (NDRC, 2010). Our spatial distribution data of check dams provide important information for optimizing the location of the new check dams in the future.

### 3.4 Uncertainty, limitation, and future work

Combined with high-resolution and easily accessible Google Earth images and object-based classification strategy, we provide the check dam dataset on the CLP for the first time. However, there are still some limitations in the extraction of check dams on the regional scale. Firstly, although this method has high accuracy and good visual effect, the whole extraction process is still semi-automatic, and visual interpretation and expert experience are still needed in the steps of



classification feature selection and human-computer interaction. Secondly, there are a few water-covered check dams on the
CLP, and the characteristics and functions of these water-covered check dams and small reservoirs are the same. It is
therefore difficult to separate water-covered check dams from reservoirs, which may have led to a slight underestimation of
the number of check dams in our dataset. Finally, due to the randomness and accessibility of historical Google Earth images,
it is difficult to obtain April and May images for the entire CLP in the same year. Considering that the large-scale ecological
restoration projects (e.g., Grain for Green project) on the CLP have significantly reduced soil erosion, the change of silted
land area of check dams in the short term is basically negligible. Therefore, we collected all available May images from 2016
to 2020 to cover more study areas. However, some newly constructed check dams may have been ignored (e.g., the acquired
image is from 2016, but the check dam was constructed in 2018), which may also lead to a slight underestimation of the
number of check dams. In future work, we can use our check dam dataset as the training set, combined with the high-
resolution and long-time series satellites (e.g., Sentinel-2) and deep learning, to propose a more automated and convenient
check dam extraction process. Additionally, we will further evaluate the carbon storage and grain supply of check dams by
combining a large amount of measured organic carbon content and grain yield data of dam land.

### 4 Data availability

The Digital elevation Model (DEM) of Chinese Loess Plateau was obtained from the Shuttle Radar Topography Mission
(SRTM) with a resolution of 30 m ([https://www.usgs.gov/centers/eros/science/usgs-eros-archive-digital-elevation-shuttle-](https://www.usgs.gov/centers/eros/science/usgs-eros-archive-digital-elevation-shuttle-)
[radar-topography-mission-srtm-1](radar-topography-mission-srtm-1)); The official statistics of the area and number of check dams in different provinces were
obtained from the Yellow River Water Resources Commission of the Ministry of Water Resources
([http://www.yrcc.gov.cn/sylm/2022stbcgb/](http://www.yrcc.gov.cn/sylm/2022stbcgb/)). The generated check dam dataset in this article is freely available at
https://doi.org/10.5281/zenodo.7619098 (Zeng et al., 2023). The vectorized dataset of check dams includes the following
attributes: longitude, latitude, dam land area, dam land perimeter, sediment volume, and sediment mass.

### 5 Conclusion
In this study, the first vectorized dataset of check dam on the CLP containing spatial distribution, silted land area, and
sediment volume of check dams was provided through object-based classification method combined with high-resolution
and easily accessible Google Earth images. The self-developed computer program combined with auxiliary data, visual
interpretation, and expert knowledge is used to improve the extraction accuracy of check dams, and finally 50,226 check
dams were extracted on the CLP. These check dams have formed 931 km$^2$ of dam land, intercepting about 10.2 billion tons
of sediment, which equals about 46% of the Yellow River's sediment load to the Bohai Sea during the period 1970-2020.
This dataset has good visual effect and high accuracy, of which the producer's accuracy and user's accuracy of the check
dam are 88.9% and 99.5%, respectively. The area and number of check dams in our dataset are highly consistent with the



latest official statistics of check dams ($R^2 > 0.99$) to further verify the accuracy of this dataset. This dataset can be further

used to quantify the contribution of check dams on the CLP to the variation of sediment load in the Yellow River, and to assess other ecosystem services functions of check dams, such as carbon sequestration and grain supply. However, to ensure the accuracy of extraction, we involved more visual interpretation and expert experience in the extraction process, which affects the extraction efficiency of check dams. Future work should focus on proposing a more automated check dam extraction method, combining the evolving high-resolution and multispectral satellites and deep learning technologies.

## Contributions


Y.Z. and N.F. conceived and designed the study. Y.Z. and T.J. wrote the paper. R.Z., B.W. and W.D. analyzed the data. B.X., J.J. and Z.S. contributed discussion. X.Y. provided assistance with the data analysis and software development. B.X., L.D., J.J. and Z.S. commented and edited the manuscript.

## Competing interests

The authors declare that they have no conflict of interest.

## Financial support

This work was supported by the National Natural Science Foundation of China (41930755, 42177335, 42207405, and U2243225).

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
