# Peer review of "Vectorized dataset of check dams on the Chinese Loess Plateau using object-based classification method from Google Earth images"

_Earth System Science Data, 2023_

## Referee Comment (RC1)

**Comments on ESSD-2023-120**

This manuscript attempts to identify the dam lands silted by check dams on the Chinese Loess Plateau using an object-based identification method in conjunction with Google Earth images featured with high spatial resolution. Moreover, the spatial distribution, dam lands area and sediment volumes were also analyzed. The results of this manuscript can provide a baseline for further studies which focus on the soil and water conservation, and grain yield on the CLP. But, there still are some serious issues in this manuscript. Firstly, the authors used a bit outdated identification method to identify the dam land on CLP. I suggest the authors try to use the Artificial Intelligence technology for example the random forest to identify the check dam and dam land. Secondly, the structure of this manuscript is not regular, for example the mixture of results and discussion. Thirdly, the writing logic and style are confused. Finally, some patches in the uploaded data are not in line with our recognition. For example, the patch is not drawn as one side with straight line which represents the check dam. This may be caused by the incorrect setting of identification rules. The detailed suggestions and comments are as follows.

P1, line 14, "invest…in doing …" is always used. So, change "to implement" to "in implementing".

P1, line 17, change "once the" to "a", and change "great" to "large".

P1, line 19, please avoid using the word "first". Change "the first" to "a".

P1, lines 21-22, change this sentence to "we first investigated and analyzed the key characteristics of check dams on the 0.3-1.0 m resolution Google Earth images during the optimum period".

P1, lines 23, insert "methods of" before "multi-scale", and change "self-developed" to "self-development".

P1, line 24, change "combined with" to "in conjunction with", and change "is" to "were".

P1, lines 25-27, change "is" to "was", and change "are" to "were".

P1, lines 28-29, insert a related phrase "not only…but also…." Before two predicates of "provides" and "will help". Delete the first comma in line 29.

P2, line 35, change "to" to "In order to".

P2, lines 39-40, suggesting delete ", such as China, Spain, Australia, America, India, Iran, and Ethiopia". Because the term of "global" has been used in first half sentence. It will confuse the reader.

P2, line 41, change "silted land" to "silt".

P2, line 42, insert "can" before "reduces". Whose runoff velocity? Soil? If true, insert "of soil" after "runoff velocity".

P2, line 42, why place a noun before the adjective (erosion kinetic)? Suggesting change "erosion kinetic energy" to "kinetic energy of erosion".

P2, line 44, change "data from different study areas" to "results at different study areas". Because "data" is a general concept. It can indicate not only the input data and output data, but also the information concluded from the tables, maps, charts and results for certain research.

P2, line 46, the check dams cannot provide the ecosystem services such as carbon sequestration and grain supply. Indeed, the ecosystem services are contributed by the silt. Please rewrite this sentence.

P2, line 41, 48, change "behind" to "intercepted by". Because the check dam is not a directional object. The direction is decided by the person's location. If you stand at the non-silt side and face to the silt, the silt is in front of the check dam. But if you stand at the silt side and face to check dam, the silt is behind the check dam.

P2, line 51, change the second "the" to "a".

P2, line 52, insert "its" before "serious".

P2, line 54, change "cooperate with" to "cooperating with".

P2, line 57, change "great" to "large", change "is" to "has been".

P2, line 58, change "how many" to "what are".

P2, line 59, suggesting give the full name of "P.R.", although most Chinese authors know it representing People's Republic.

P2, line 60, insert "they" before "captured".

P2, lines 61-62, change "remained intact till" to "in".

P2, line 62, change "silt sediment" to "sediment volume".

P2, line 63, change "data" to "amounts".

P3, line 66, change "between" to "from", change "detail" to "detailed".

P3, line 67, change "remains unclear" to "remain unrecorded".

P3, line 68, delete "data".

P3, line 70, change "benefits of" to "benefitting from".

P3, line 72, change "incalculable" to "some" or "a few". "incalculable" is an adjective, which cannot modify adjective. Moreover, the mood of "incalculable" is too absolute. Please be careful when using these words. If you using this word at here, it means the check dams amount is not creditable, which will weaken the credibility of the government.

P3, line 73, change "has made" to "has been made". Because the subject of this sentence is "remote sensing". Change "in" to "in fields of".

P3, lines 74-76, the mood of this sentence is too absolute. There are no limited phrases in this sentence. These two "studies" are just your findings through reading published articles. However, whether you have read all the related articles in the world about this topic? Or writing in other languages (not just in Chinese and English)? Moreover, two articles (Li et al., 2021; Tian et al., 2013) were cited at end of this sentence, but the next sentence was started with Zhao et al. (2013). Obviously, there are three articles related with the topic of check dam exploration using remote sensing technology. Please rewrite this sentence.

P3, line 77, which sensor's images were implied in Zhao et al. (2013)? Landsat 5? Landsat 7?

P3, line 80, the deep learning and object-based classification can not be applied into the identification of check dams at a larger study area? In fact, the identification effect using machine learning is decided by the input data and specified algorithm. The study area is not a key factor affecting the identification effect. Moreover, these words of "first" and "blank" should be avoided in the article. That is to say, we should always address a sentence based on the objective fact, not the subjective assumption when we write a manuscript. Please rewrite these two sentences.

P3, line 81, delete the second "the".

P3, line 82, change "corresponding" to "correspondingly".

P3, line 82, suggesting transfer "hm$^2$" to "km$^2$".

P3, line 83, change "product" to "generate". Suggesting change this sentence to "The decametric-resolution images (e.g., Landsat-7/8) may generate erroneous judgement for identification of silt due to the mixed pixels".

P3, line 84, change "silted land formed" to "silt intercepted".

P3, line 85, change "distinguish" to "be distinguished". What is slope cropland? What is the difference between slope cropland and terrace? Are terraces used for planting crops? So, change "slope cropland and terrace" to "cropland".

P3, line 87, change "identify" to "be identified", and change "extraction" to "identified".

P3, line 88, change "focus on" to "impacting on", delete the first "and", change "within the category" to "characteristics", change "correlation" to "correlative". Change "consider" to "refer to".

P3, line 89, change "this method also produces" to "these methods also emerge".

P3, line 90, change "integrity" to "accuracy".

P3, line 91, delete "comprehensively" and "statistical".

P3, line 91, change "considers" to "referring to". Because the subject of "consider" is sb, not sth.

P3, line 92, delete "and", and change "silted land extraction results combined with high-resolution" to "identified results combing with high spatial resolution".

P3, lines 93-94, change this sentence to "Therefore, we attempt to identify the check dams on the CLP using object-based classification method in conjunction with high spatial resolution (0.3-1.0 m) and easily accessible Google Earth images in this study".

P3, line 94, change "self-developed" to "self-development".

P3, line 95, change "combined with" to "in conjunction with", change "is" to "are", change "extraction" to "identified".

P3, line 96, change "test" to "testing", change "This dataset" to "This study"

P3, line 97, delete "function", change "to provide" to "offers"

P4, line 100, suggesting change "extraction" or "extract" to "identification" or "identify" through this manuscript.

P4, lines 100-101, suggesting change this sentence to "It is noticeable that the dam land mentioned in the study mean the eroded sediment captured by the check dam rather than the dam body".

P4, lines 101-102, suggesting change this sentence to "Because the functions of check dams such as sediment retention, carbon sequestration, and grain supply are embodied by the dam land".

P4, lines 104-106, who's efficiency? Improve identified efficiency of dam land? Change "concentrated" to "dense". The spatial distribution characteristics of check dams was concluded from the dataset of Liu et al., 2021a or the dataset manufactured in this study? Change "divide" to "divided".

P4, line 106, change "concentrated" to "dense".

P4, line 108, why the acquisition and processing of corresponding Google Earth images in dam sparse region are complex? Does the large area is the only reason? If you use some auto download software and some auto-processing script for the Google Earth images, I think the workload is not an obstacle in you study.

P4, lines 107-109, these two sentences can be merged into one sentence. For example, "However, owning to the large area but with just about 15% of the check dams in the dam sparse region, the acquisition and processing of corresponding Google Earth images is time-consuming."

P4, line 110, do you directly identify the dam land in software of Google Earth? Is there no need to download the images? How do you ensure the consistency of dam land using two different identified methods? Change "Noticeable" to "Noticeably".

P4, line 110, the authors listed three regions without check dams, but we found that they were not mapped on the Fig. 2a. Please remap that sub-figure. If you do that, the readers can quickly locate the positions of three regions on the sub-figure.

P4, line 111, change the comma to period (full stop) and insert a comma after "so", change "so" to "So".

P4, lines 110-112, these two sentences can be merged into one sentence. For example, "Noticeably, we masked three regions without check dams such as the Mu Us Desert in the northwest of CLP, the Guanzhong Plain in the middle of CLP, and the Rocky Mountains in the east of CLP, which has significantly reduced the workload of visual interpretation".

P4, line 112, change "aggregate" to "aggregated". Change this sentence to "Finally, we aggregated these two dam land datasets and verified their accuracy through…..". the sequent word of "finally" was suddenly appeared in the last sentence of this paragraph, but we cannot find other sequent words such as "firstly", "secondly" in this paragraph.

P4, line 115, please adjust the workflow in this study. Please assign step 7 to Dam sparse region and reassign step 8 to Accuracy verification.

P4, line 117, delete "vast", change "cultivated" to "cultivatable".

P4, line 118, change "of" to "in", change "cultivated" to "cultivatable", and delete "dams".

P4, line 119, what does "abandoned dam land" mean? How to distinguish cultivable dam land and abandoned dam land?

P4, line 119, insert "lived" before "on", and insert "each year" after "May".

P5, line 120, this sentence needs to be questioned. Because in Northern China, the growing of vegetation is not luxuriant in May. So, it will result in misjudgment for land cover type.

P5, line 121, change "from" to "on".

P5, lines 121-122, this sentence can be readdressed as "The distinguishable difference can be confirmed by the field investigations". Actually, the input data is the Google Earth images, you cannot find the similar or difference through comparing the images from same source although they were

achieved during different phases. Comparing Google Earth images with field photographs is feasible.

P5, line 123, change "$N_{dam}$" to "$N_{sur}$".

P5, lines 123-124, in the brackets, three counties are Baota, Zizhou and Lin, but on the figure 3a, three counties are Zizhou, Linxian and Yan'an. Please check the names and make sure the writing of English name is correct (do not mix Chinese and English, for example, xian-county).

P5, lines 124-125, why do the authors select the images in 2019 to calculate NDVI? From the figure 2, we can see that the photographs were taken in May 2018. Moreover, the NDVI was calculated by Google Earth images or Sentinel-2 images or both? Finally, if the authors calculated the NDVI in Google Earth Engine, how do you deal with the noises for example cloud?

P5, line 125, insert "values" before "of", change "are" to "were".

P5, line 126, change "Fig .3" to "Fig .3b-d" and insert ", 2019" after "May".

P5, line 127, what does "May images" mean? Does it mean the images in May? If so, move "May" before "2016" and "2020". Moreover, do all available images mean only the images from 2016 to 2020 are available? Can the images before 2016 and after 2020 be available? delete "a".

P5, line 128, change "taken" to "retrieved".

P5, line 129, change "satellites" to "sensors".

P5, lines 129-130, change the last half sentence to "which may result in potential chromatic aberration at the junction of two image scenes".

P5, lines 130-131, this sentence confused me a little. Do the authors mean the images were regrouped according to the launching date of satellite sensors? And finally, the authors obtained 52 scenes of image in total.

P5, Figure 2a, P6, Figure 3a, where does the dam lands data come from? From this study or other resource? If it comes from this study, the logic of manuscript writing is reverse. If it comes from other source, please cite the source in the caption of figures. Moreover, suggesting assign the same color to dam lands on two sub-figures. Suggesting copy the same administrative divisions map of figure 2a to figure 3a. meanwhile, suggesting copy the same topographical map of figure 3a to figure 2a. Delete the graticules on

sub-figures 2a, 3a, because the authors have plotted the North arrow on these two sub-figures.

P5, Figure 2b-d, suggesting address the date with the format "MM DD YYYY".

P6, Figure 3b-d, suggesting plot the axis (both x-axis and y-axis) ticks, so the readers can quicky find the date range filled by blue rectangle on the x-axis.

P6, line 141, insert the version and company of software after "eCognition Developer".

P6, line 144, delete the first "parameters".

P6, line 148, change "extraction" to "identification".

P6, line 149, delete the two "the", delete "parameters", change "classification" to "identification".

P6, line 150, how do the authors make combinations between shape and compactness? The authors should give a simple description at here.

P7, line 157, change "classification" to "identification".

P8, line 158, change "assign" to "assigned".

P8, line 159, delete "conditions".

P8, line 160, change "some features", "assign" and "classification" to "a feature", "assigned" and "identification".

P8, line 161, change "classification" to "identification", because in this study, only dam land should be identified from images. Change "was" to "is". Do the authors mean "as our goal is to identify more dam land in a target category"?

P8, line 162, change "first" to "firstly", change two "land" to "lands".

P8, line 163, change "size" to "amplitude", change "thresholds algorithm" to "threshold algorithms".

P8, line 164, change "classification" to "identification".

P8, line 165, delete the second "the".

P8, line 166, change "combined with" to "using".

P8, line 167, change "land was" to "lands were". What does "included in the classification range" mean? It confuses me a lot. Change "of" to "in".

P8. Line 168, change "concentrated" to "dense", change "mainly" to "main", change "the cropland on slope land" to "the ambient croplands".

P8, line 169, change "superposition" to "overlay".

P8, line 171, give the full name of "SRTM-DEM", insert "spatial" before "resolution", insert the version and company of software after "ArcGIS".

P8, line 172, change "superimpose the river network and the bare land layer to extract" to "overlaid the river network layer on the bare land layer to identify".

P8, line 173, insert "identified" before "accuracy", delete "extraction".

P8, line 174, change "classification" to "identification".

P8, lines 174-175, this sentence can be addressed as "The left window with showing high-resolution Google Earth image can link with the right window with representing the dam lands layer identified on the previous steps".

P8, line 175, change "assign" to "assigned", change "extracted" to "identified".

P8, lines 176-178, change this sentence to "Finally, we merged the vector polygons identified from 52 images with the attribute value of 1 in ArcGIS".

P9, line 183, change "take" to "took".

P9, line 183, as addressed in the Introduction, two studies (Li et al., 2021; Tian et al., 2013) have explored the identification of check dam on Chinese Loess Plateau. Moreover, whether the latitudes and longitudes of check dams published by Ministry of Water Resource of the People's Republic of China are available? So, the authors cannot be addressed as "there is no available spatial distribution dataset of check dams on the CLP".

P9, line 184, which dataset should be verified? Check dams or dam lands?

P9, line 185, change "traditional" to "traditionally".

P9, line 186, change "determine" to "determined". Which dataset was taken as reference? The number of check dams published by Ministry of Water Resource of the People's Republic of China?

P9, line 187, change "concentrated" to "dense".

P10, lines 188-189, it confuses me a lot. What did the samples be used to verify? Check dams or dam lands?

P10, line 189, delete the second and third "the".

P10, line 190, insert "accuracy" before "(PA)".

P10, line 192, change "compare" to "compared", change "official" to "officially".

P10, line 194, change "check dam" to "dam land".

P10, line 199, there is a cited error for Zeng et al. (2022). Zeng et al. (2022a) or Zeng et al. (2022b)? delete "method".

P10, line 201, why not use the general unit of "$km^2$"?

P10, line 205, change "in which" to "and", change "check dam map" to "dam land".

P10, line 206, delete "class".

P10, line 208, change "maps" to "map", change "2000" to "1947".

P11, line 210, change "combined with" to "used".

P11, line 213, delete "a slope of 1.1185 and".

P11, lines 213-215, suggesting change this sentence to "The number of check dams in this study is closer to that in official statistics reported in 2013 (58, 446) rather than that reported in 2003 (110, 000)".

P11, line 217, change "statistical" to "statistic" of the labels on the y-axis.

P11, line 218, please revise the caption of figure 6. For example, Area (a) and number (b) comparison of check dams between this study and official statistic.

P12, line 223, what does the significance of figure 7? Do the 12 sub-images represent the check dams in typical region or with special feature?

P12, line 226, change "number of" to "total", delete "are".

P12, line 229, what does "which" indicate? The spatial distribution of check dams or regional soil erosion? Suggesting split this long sentence into two short sentences.

P12, line 230, change the second "in" to "cited from", and insert the citation after "GBT16453.3-1996".

P12, line 231, delete the second "the", change "ranges from" to "of".

P12, line 232, transfer the unit of "$hm^2$" to "$km^2$", insert a comma after "2, 250".

P13, line 233, change "the highest" to "high".

P13, line 234, delete "number of", change "basically" to "almost", change "the results previously reported" to "the previously reported results".

P13, line 236, change "in different regions" to "at provincial level".

P13, line 237, change "Region" to "Province", change "Area" to "Silted area", change "Volume" to "Sediment volume".

P13, line 239, change "account for" to "accounting for", change "billion" to "$\times 10^9$".

P13, line 240, change "estimated" to "calculated", change "previously reported" to "in previous report".

P13, line 241, insert "on the CLP" before "intercept", change "billion" to "$\times 10^9$", change "to" to "into".

P13, line 242, delete "the period".

P13, line 243, suggesting change the x-labels with legend labels. That is to say, the x-labels are listed as Micro, Small, Medium, and Large, but legend label are listed as 0-0.2 $km^2$, 0.2-2 $km^2$, 2-7 $km^2$, >7 $km^2$.

P14, line 249, change "role" to "roles".

P14, line 250, change "of" to "among". "to date" is always located at the head or end of a sentence.

P14, line 251, change "laborious" to "laboriously", change "find check dams suitable for the research objectives" to "find suitable check dams satisfied with the research objectives".

P14, line 252, change "precise" to "precisely".

P14, line 253, change the first "of" to "for", change "provides" to "also offers".

P14, line 254, change ", which are crucial for those studies using" to "for those studies which use".

P14, lines 256-257, do not use the phrase "fill the gap" and the word "unprecedented". The authors should use euphemism in the manuscript as far as possible. Please revise this sentence.

P14, line 260, change "billion" to "$\times 10^9$".

P14, line 261, delete "the period".

P14, lines 261-262, change "which equals 46% of the sediment load of the Yellow River, once the largest sediment contributor to the global ocean" to "equaling 46% of the sediment load of the Yellow River, which was once the largest sediment contributor to the global ocean".

P14, line 263, change "unprecedented" to "enormous", delete "reduction".

P14, lines 263-265, change "studies" to "article", change "emphasize" to "emphasizes", change "point" to "points", because only one citation was listed in the brackets at end of this sentence. Change "potential" to "potentiality", delete "important".

P14, line 266, change "large scale" to "macro-scale", change "or" to "with". Does literature compilation mean meta-analysis?

P14, lines 267-268, change this sentence to "Moreover, the soil silted by the check dams has higher moisture than that of terrace and slope cropland".

P14, line 268, change "shows" to "showed".

P14, line 270, change "determine" to "estimate".

P14, line 271, it is unbelievable that the Chinese government plans to build 56161 new check dams on the CLP till 2030. Because, as shown in this

study, there are 50226 check dams on the CLP from 1970s to 2018. I cannot believe the government has so ambitious planning to build the nearly same check dams as that in the past 50 years during the short term of future seven years. I do not think the more check dams are good for local ecosystem. It even will break the balance between agricultural land and ecological land.

P14, line 272, change "provide" to "provides".

P14, line 275, change "Combined" to "combining".

P14, line 276, delete "for the first time", change "extraction" to "identification".

P14, line 277, delete the first "the".

P14, line 278, change "extraction" to "identification".

P15, line 280, change this sentence to "and their characteristics and functions are same as small reservoirs".

P15, line 281, change "separate" to "distinguish", change "have led to" to "lead to".

P15, line 283, change this sentence to "it is difficult to obtain the images for the entire CLP in April and May of the same year".

P15, lines 285-286, change this sentence to "Therefore, we collected all available images covering most study areas in May from 2016 to 2020".

P15, lines 286-287, change the sentence in the brackets to "e.g., the check dam was constructed in 2018, but it was not imaged in 2016".

P15, line 288, change "combined" to "combing".

P15, line 289, the Sentinel-2A was launched at 23 June, 2015. Its accessible images were not earlier than that of Quickbird. Moreover, the highest spatial resolution of Sentinel-2A is 10 m, which is far lower than that of Quickbird (0.3-1.0 m). Its most advantage is for free downloading. So, if the authors want to achieve check dam dataset with higher spatial resolution, the Quickbird images are the best choice, although you should pay much more fee on them.

P15, line 290, change "extraction process" to "identification script".

P15, line 293, change the initial letter of "Digital" and "Model" to lowercase.

P15, line 301, delete "first".

P15, line 302, delete "of check dams", change "provided" to "manufactured", change "classification" to "identification", change "combined with" to "in conjunction with".

P15, line 303, change "self-developed" to "self-development".

P15, line 304, change "extraction" to "identification".

P15, line 305, change "extracted" to "identified", change "billion" to "$\times 10^9$"

P15, line 306, change "to" to "into", delete "the period".

P15, line 309, delete "to further verify the accuracy of this dataset".

P15, line 312, change "accuracy of extraction" to "identified accuracy", change "extraction" to "identification".

P15, line 313, change "extraction" to "identification".

P15, line 314, change "extraction" to "identification".

---

## Author Comment (AC1)

**Reviewer #1:**

Firstly, the authors used a bit outdated identification method to identify the dam land on CLP. I suggest the authors try to use the Artificial Intelligence technology for example the random forest to identify the check dam and dam land.

Response: Thank you for your comment. We have pointed out that the method employed in this study remains semi-automatic (Section 3.4 Limitation). However, when integrated with our self-developed program for human-machine interaction identification, this method demonstrates high accuracy and produces favourable visual effects. Although artificial intelligence technology exhibits high efficiency, it may also lead to more omission and commission errors. We believe that *ensuring accuracy is even more important* for the basic dataset provided for the first time, so we have added more visual interpretation and expert experience in the steps of identification feature selection and human-computer interaction. Subsequently, we will utilize our check dam dataset as the training set, in conjunction with high-resolution satellites equipped with additional multispectral bands (e.g., Sentinel-2) and deep learning techniques, to propose an enhanced, automated, and convenient process for check dam identification in future work (Section 3.4 future work).

Secondly, the structure of this manuscript is not regular, for example the mixture of results and discussion.

Response: Thank you for your comment. We have referred to the writing structure of some similar topic studies published in the Earth System Science Data journal, which merged the results section and the discussion section.

References:

1. Yang J, Huang X. The 30 m annual land cover dataset and its dynamics in China from 1990 to 2019[J]. Earth System Science Data, 2021, 13(8): 3907-3925.

2. Cao B, Yu L, Naipal V, et al. A 30 m terrace mapping in China using Landsat 8 imagery and digital elevation model based on the Google Earth Engine[J]. Earth

System Science Data, 2021, 13(5): 2437-2456.

3.  Li B, Xu X, Liu X, et al. An improved global land cover mapping in 2015 with 30 m resolution (GLC-2015) based on a multi-source product fusion approach[J]. Earth System Science Data, 2022: 1-35.

Thirdly, the writing logic and style are confused.

Response: We express our sincere appreciation for your thorough review and valuable revision suggestions, which have significantly enhanced the quality of our manuscript. We have made detailed modifications throughout the entire manuscript according to your suggestions.

Finally, some patches in the uploaded data are not in line with our recognition. For example, the patch is not drawn as one side with straight line which represents the check dam. This may be caused by the incorrect setting of identification rules.

Response: Thank you for your comment. In some regions, the utilization of multiresolution segmentation may encounter challenges associated with image quality, leading to irregular edges or erroneous classification, which is difficult to avoid. Overall, our identification of dam lands remains high accuracy and good visual effect (Figure 2b-d and Figure 7).

[Figure]

**Figure 2: Check dams in the study area. (a) Dam concentration region on the Chinese Loess Plateau, (b-d) Google Earth images, photographic images, and unmanned aerial vehicle images of dam lands in May.**

[Figure]

**Figure 7: Spatial distribution of check dams on the Chinese Loess Plateau.**

Note: To avoid unnecessary redundancy that may affect reviewers' and readers' viewing, we did not provide point-to-point responses to grammar-related modifications. All grammar issues have been rectified based on the reviewer's suggestions.

1. P1, line 14, "invest…in doing …" is always used. So, change "to implement" to "in implementing".

2. P1, line 17, change "once the" to "a", and change "great" to "large".

P1, line 19, please avoid using the word "first". Change "the first" to "a".

P1, lines 21-22, change this sentence to "we first investigated and analyzed the key characteristics of check dams on the 0.3-1.0 m resolution Google Earth images during the optimum period".

P1, lines 23, insert "methods of" before "multi-scale", and change "self-developed" to "self-development".

P1, line 24, change "combined with" to "in conjunction with", and change "is" to "were".

P1, lines 25-27, change "is" to "was", and change "are" to "were".

P1, lines 28-29, insert a related phrase "not only…but also…." Before two predicates of "provides" and "will help". Delete the first comma in line 29.

P2, line 35, change "to" to "In order to".

P2, lines 39-40, suggesting delete ", such as China, Spain, Australia, America, India, Iran, and Ethiopia". Because the term of "global" has been used in first half sentence. It will confuse the reader.

P2, line 41, change "silted land" to "silt".

Response: Thank you for your suggestion. Check dam forms a special land use by intercepting eroded sediment, which is usually called "silted land" or "dam land" in the literature. We believe that incorporating the term "land" provides a more accurate description of the object we are identifying.

P2, line 42, insert "can" before "reduces". Whose runoff velocity? Soil? If true, insert "of soil" after "runoff velocity".

Response: Thank you for your comment. We have changed "runoff velocity" to "surface runoff velocity".

P2, line 42, why place a noun before the adjective (erosion kinetic)? Suggesting change "erosion kinetic energy" to "kinetic energy of erosion".

P2, line 44, change "data from different study areas" to "results at different study areas". Because "data" is a general concept. It can indicate not only the input data and output data, but also the information concluded from the tables, maps, charts and results for certain research.

P2, line 46, the check dams cannot provide the ecosystem services such as carbon sequestration and grain supply. Indeed, the ecosystem services are contributed by the silt. Please rewrite this sentence.

Response: Thank you for your comment. It is precisely because of the construction of check dams in the gully that the silt and associated organic carbon can be intercepted and buried, and subsequently used for planting crops. We have changed this sentence to "In addition to preventing soil erosion, the construction of check dams also provides more unexpected ecosystem services, including carbon sequestration and grain supply." (Lines 49-50 in Tracked Changes, same below)

P2, line 41, 48, change "behind" to "intercepted by". Because the check dam is not a directional object. The direction is decided by the person's location. If you stand at the non-silt side and face to the silt, the silt is in front of the check dam. But if you stand at the silt side and face to check dam, the silt is behind the check dam.

P2, line 51, change the second "the" to "a".

P2, line 52, insert "its" before "serious".

P2, line 54, change "cooperate with" to "cooperating with".

P2, line 57, change "great" to "large", change "is" to "has been".

P2, line 58, change "how many" to "what are".

P2, line 59, suggesting give the full name of "P.R.", although most Chinese authors

know it representing People's Republic.

P2, line 60, insert "they" before "captured".

P2, lines 61-62, change "remained intact till" to "in".

P2, line 62, change "silt sediment" to "sediment volume".

P2, line 63, change "data" to "amounts".

P3, line 66, change "between" to "from", change "detail" to "detailed".

P3, line 67, change "remains unclear" to "remain unrecorded".

P3, line 68, delete "data".

P3, line 70, change "benefits of" to "benefitting from".

P3, line 72, change "incalculable" to "some" or "a few". "incalculable" is an adjective, which cannot modify adjective. Moreover, the mood of "incalculable" is too absolute. Please be careful when using these words. If you using this word at here, it means the check dams amount is not creditable, which will weaken the credibility of the government.

P3, line 73, change "has made" to "has been made". Because the subject of this sentence is "remote sensing". Change "in" to "in fields of".

P3, lines 74-76, the mood of this sentence is too absolute. There are no limited phrases in this sentence. These two "studies" are just your findings through reading published articles. However, whether you have read all the related articles in the world about this topic? Or writing in other languages (not just in Chinese and English)? Moreover, two articles (Li et al., 2021; Tian et al., 2013) were cited at end of this sentence, but the next sentence was started with Zhao et al. (2013). Obviously, there are three articles related with the topic of check dam exploration using remote sensing technology. Please rewrite this sentence.

Response: We have changed "two studies" to "a few studies".

P3, line 77, which sensor's images were implied in Zhao et al. (2013)? Landsat 5? Landsat 7?

Response: We have changed " Landsat images " to " Landsat-5 TM images".

P3, line 80, the deep learning and object-based classification can not be applied into the identification of check dams at a larger study area? In fact, the identification effect using machine learning is decided by the input data and specified algorithm. The study area is not a key factor affecting the identification effect. Moreover, these words of "first" and "blank" should be avoided in the article. That is to say, we should always address a sentence based on the objective fact, not the subjective assumption when we write a manuscript. Please rewrite these two sentences.

Response: Thank you for your comment. We have changed these two sentences to "Nevertheless, these studies have only explored different methods to extract check dams on a very small scale, and did not be extended to the whole CLP. That is, the current dataset of check dams on the CLP is still lacking." (Lines 83-84)

P3, line 81, delete the second "the".

P3, line 82, change "corresponding" to "correspondingly".

P3, line 82, suggesting transfer "hm2" to "km2".

Response: Thank you for your suggestion. Converting units to "km$^2$" will result in too many decimal places.

P3, line 83, change "product" to "generate". Suggesting change this sentence to "The decametric-resolution images (e.g., Landsat-7/8) may generate erroneous judgement for identification of silt due to the mixed pixels".

P3, line 84, change "silted land formed" to "silt intercepted".

P3, line 85, change "distinguish" to "be distinguished". What is slope cropland? What is the difference between slope cropland and terrace? Are terraces used for planting crops? So, change "slope cropland and terrace" to "cropland".

Response: The following figure clearly shows the difference between slope cropland and terrace. Slope cropland usually refers to cropland with a slope of 6-25°, with low crop yield and severe soil erosion.

[Figure]

P3, line 87, change "identify" to "be identified", and change "extraction" to "identified".

P3, line 88, change "focus on" to "impacting on", delete the first "and", change "within the category" to "characteristics", change "correlation" to "correlative". Change "consider" to "refer to".

P3, line 89, change "this method also produces" to "these methods also emerge".

P3, line 90, change "integrity" to "accuracy".

P3, line 91, delete "comprehensively" and "statistical".

P3, line 91, change "considers" to "referring to". Because the subject of "consider" is sb, not sth.

P3, line 92, delete "and", and change "silted land extraction results combined with high-resolution" to "identified results combing with high spatial resolution".

P3, lines 93-94, change this sentence to "Therefore, we attempt to identify the check dams on the CLP using object-based classification method in conjunction with high spatial resolution (0.3-1.0 m) and easily accessible Google Earth images in this study".

P3, line 94, change "self-developed" to "self-development".

P3, line 95, change "combined with" to "in conjunction with", change "is" to "are", change "extraction" to "identified".

P3, line 96, change "test" to "testing", change "This dataset" to "This study"

P3, line 97, delete "function", change "to provide" to "offers"

P4, line 100, suggesting change "extraction" or "extract" to "identification" or "identify" through this manuscript.

P4, lines 100-101, suggesting change this sentence to "It is noticeable that the dam land mentioned in the study mean the eroded sediment captured by the check dam rather than the dam body".

P4, lines 101-102, suggesting change this sentence to "Because the functions of check dams such as sediment retention, carbon sequestration, and grain supply are embodied by the dam land".

P4, lines 104-106, who's efficiency? Improve identified efficiency of dam land? Change "concentrated" to "dense". The spatial distribution characteristics of check dams was concluded from the dataset of Liu et al., 2021a or the dataset manufactured in this study? Change "divide" to "divided".

Response: Thank you for your comment. The spatial distribution characteristics of check dams was concluded from the dataset manufactured in this study.

P4, line 106, change "concentrated" to "dense".

P4, line 108, why the acquisition and processing of corresponding Google Earth images in dam sparse region are complex? Does the large area is the only reason? If you use some auto download software and some auto-processing script for the Google Earth images, I think the workload is not an obstacle in you study.

Response: Thank you for your comment. Downloading Google Earth images is not a difficult task. Each image requires a series of intricate procedures, encompassing processes such as multiresolution segmentation, assign class, and human-computer interaction. These intricate workflows prove immensely challenging for the dam sparse region spanning an expanse of 438,000 km$^2$.

P4, lines 107-109, these two sentences can be merged into one sentence. For example, "However, owning to the large area but with just about 15% of the check dams in the dam sparse region, the acquisition and processing of corresponding Google Earth

images is time-consuming."

P4, line 110, do you directly identify the dam land in software of Google Earth? Is there no need to download the images? How do you ensure the consistency of dam land using two different identified methods? Change "Noticeable" to "Noticeably".

Response: Thank you for your comment. We employed the polygon tool on Google Earth, in conjunction with artificial visual interpretation, to identify the dam land in the dam sparse area. For the dam dense region, we also used artificial visual interpretation to further improve the accuracy of dam land identification after the object-based classification. Our ultimate objective is to furnish a vectorized dataset of check dams that can be utilized by policymakers and scientific practitioners. Although artificial visual interpretation may impact identification efficiency, it will significantly enhance precision.

P4, line 110, the authors listed three regions without check dams, but we found that they were not mapped on the Fig. 2a. Please remap that sub-figure. If you do that, the readers can quickly locate the positions of three regions on the sub-figure.

Response: Thank you for your suggestion. There are still a few check dams in these three regions. In order to avoid unnecessary misunderstanding, we deleted this sentence in the manuscript. (Lines 123-125)

P4, line 111, change the comma to period (full stop) and insert a comma after "so", change "so" to "So".

P4, lines 110-112, these two sentences can be merged into one sentence. For example, "Noticeably, we masked three regions without check dams such as the Mu Us Desert in the northwest of CLP, the Guanzhong Plain in the middle of CLP, and the Rocky Mountains in the east of CLP, which has significantly reduced the workload of visual interpretation".

P4, line 112, change "aggregate" to "aggregated". Change this sentence to "Finally, we aggregated these two dam land datasets and verified their accuracy through…..". the sequent word of "finally" was suddenly appeared in the last sentence of this paragraph, but we cannot find other sequent words such as "firstly", "secondly" in this paragraph.

Response: Thank you for your comment. We have changed this sentence to "We then aggregated the dam land layers of these two regions and verify the accuracy with two different validation sets. "(Lines 125-126)

P4, line 115, please adjust the workflow in this study. Please assign step 7 to Dam sparse region and reassign step 8 to Accuracy verification.

Response: Thank you for your suggestion. We have adjusted the workflow in this study to assign step 7 to dam sparse region and reassign step 8 to accuracy verification.

P4, line 117, delete "vast", change "cultivated" to "cultivatable".

P4, line 118, change "of" to "in", change "cultivated" to "cultivatable", and delete "dams".

P4, line 119, what does "abandoned dam land" mean? How to distinguish cultivable dam land and abandoned dam land?

Response: Abandoned dam land usually refers to dam land that has been abandoned and is no longer used for crop cultivation. These two different types of dam land can be distinguished by field investigation or satellite images to observe whether they are plowed during the plowing period (The Loess Plateau is usually around May).

P4, line 119, insert "lived" before "on", and insert "each year" after "May".

P5, line 120, this sentence needs to be questioned. Because in Northern China, the growing of vegetation is not luxuriant in May. So, it will result in misjudgment for land cover type.

Response: A large number of field investigations and Google Earth image observations (see the figure below) can confirm that although the vegetation is not luxuriant, it is enough to distinguish it from the dam land.

[Figure]

P5, line 121, change "from" to "on".

P5, lines 121-122, this sentence can be readdressed as "The distinguishable difference can be confirmed by the field investigations". Actually, the input data is the Google Earth images, you cannot find the similar or difference through comparing the images from same source although they were achieved during different phases. Comparing Google Earth images with field photographs is feasible.

P5, line 123, change "Ndam" to "Nsur".

P5, lines 123-124, in the brackets, three counties are Baota, Zizhou and Lin, but on the figure 3a, three counties are Zizhou, Linxian and Yan'an. Please check the names and make sure the writing of English name is correct (do not mix Chinese and English, for example, xian-county).

P5, lines 124-125, why do the authors select the images in 2019 to calculate NDVI? From the figure 2, we can see that the photographs were taken in May 2018. Moreover,

the NDVI was calculated by Google Earth images or Sentinel-2 images or both? Finally, if the authors calculated the NDVI in Google Earth Engine, how do you deal with the noises for example cloud?

Response: Thank you for your comment. Because there were a large number of clouds in the Sentinel-2 images of these three counties around May 2018. Considering that the large-scale ecological restoration projects (e.g., Grain for Green project) on the CLP have significantly reduced soil erosion, the change of silted land area of check dams in the short term (from 2018 to 2019) is basically negligible. We calculated NDVI based on Sentinel-2 images in Google Earth Engine. We used the following code to perform masking clouds and shadows in Google Earth Engine:

```
function mask Cloud and Shadows(image) {
    var cloudProb = image.select('MSK_CLDPRB');
    var snowProb = image.select('MSK_SNWPRB');
    var cloud = cloudProb.lt(5);
    var snow = snowProb.lt(5);
    var scl = image.select('SCL');
    var shadow = scl.eq(3); // 3 = cloud shadow
    var cirrus = scl.eq(10); // 10 = cirrus
    // Cloud probability less than 5% or cloud shadow classification
    var mask = (cloud.and(snow)).and(cirrus.neq(1)).and(shadow.neq(1));
    return image.updateMask(mask);
}
```

P5, line 125, insert "values" before "of", change "are" to "were".

P5, line 126, change "Fig .3" to "Fig .3b-d" and insert ", 2019" after "May".

P5, line 127, what does "May images" mean? Does it mean the images in May? If so, move "May" before "2016" and "2020". Moreover, do all available images mean only the images from 2016 to 2020 are available? Can the images before 2016 and after 2020 be available? delete "a".

Response: A significant change in the dam land area can occur when the time span is

extensive, resulting in a higher margin of error in identifying the dam land. This work started around 2018, so we chose the images from 2016 to 2020.

P5, line 128, change "taken" to "retrieved".

P5, line 129, change "satellites" to "sensors".

P5, lines 129-130, change the last half sentence to "which may result in potential chromatic aberration at the junction of two image scenes".

P5, lines 130-131, this sentence confused me a little. Do the authors mean the images were regrouped according to the launching date of satellite sensors? And finally, the authors obtained 52 scenes of image in total.

Response: The images were regrouped according to the launching date of satellite sensors, and finally we obtained 52 scenes of images in total. We conducted subsequent object-based classification for each scene image.

P5, Figure 2a, P6, Figure 3a, where does the dam lands data come from? From this study or other resource? If it comes from this study, the logic of manuscript writing is reverse. If it comes from other source, please cite the source in the caption of figures. Moreover, suggesting assign the same color to dam lands on two sub-figures. Suggesting copy the same administrative divisions map of figure 2a to figure 3a. meanwhile, suggesting copy the same topographical map of figure 3a to figure 2a. Delete the graticules on sub-figures 2a, 3a, because the authors have plotted the North arrow on these two sub-figures.

Response: The dam lands data come from this study. In order to enhance readers' intuitive understanding of the spatial distribution of dam lands, as well as their corresponding satellite images and field photos (Figure 2-d), we incorporated the check dams' spatial distribution data into the study area (Figure 2a). We attempted to incorporate the administrative boundary, terrain, and distribution of check dams as per your suggestion. However, the excessive number of elements in the figure resulted in an unattractive graph. We have unified the colour of the dam lands on two sub-figures and removed the north arrow.

P5, Figure 2b-d, suggesting address the date with the format "MM DD YYYY".

P6, Figure 3b-d, suggesting plot the axis (both x-axis and y-axis) ticks, so the readers can quicky find the date range filled by blue rectangle on the x-axis.

Response: We have made modifications to Figures 2 and 3 according to your suggestions.

P6, line 141, insert the version and company of software after "eCognition Developer".

P6, line 144, delete the first "parameters".

P6, line 148, change "extraction" to "identification".

P6, line 149, delete the two "the", delete "parameters", change "classification" to "identification".

P6, line 150, how do the authors make combinations between shape and compactness? The authors should give a simple description at here.

Response: We have added a sentence later to describe: "Therefore, we made different combinations of these parameters to test the best parameter settings in combination with visual inspection." (Lines 170-172)

P7, line 157, change "classification" to "identification".

P8, line 158, change "assign" to "assigned".

P8, line 159, delete "conditions".

P8, line 160, change "some features", "assign" and "classification" to "a feature", "assigned" and "identification".

P8, line 161, change "classification" to "identification", because in this study, only dam land should be identified from images. Change "was" to "is". Do the authors mean "as our goal is to identify more dam land in a target category"?

P8, line 162, change "first" to "firstly", change two "land" to "lands".

P8, line 163, change "size" to "amplitude", change "thresholds algorithm" to "threshold algorithms".

P8, line 164, change "classification" to "identification".

P8, line 165, delete the second "the".

P8, line 166, change "combined with" to "using".

P8, line 167, change "land was" to "lands were". What does "included in the classification range" mean? It confuses me a lot. Change "of" to "in".

P8. Line 168, change "concentrated" to "dense", change "mainly" to "main", change "the cropland on slope land" to "the ambient croplands".

P8, line 169, change "superposition" to "overlay".

P8, line 171, give the full name of "SRTM-DEM", insert "spatial" before "resolution", insert the version and company of software after "ArcGIS".

P8, line 172, change "superimpose the river network and the bare land layer to extract" to "overlaid the river network layer on the bare land layer to identify".

P8, line 173, insert "identified" before "accuracy", delete "extraction".

P8, line 174, change "classification" to "identification".

P8, lines 174-175, this sentence can be addressed as "The left window with showing high-resolution Google Earth image can link with the right window with representing the dam lands layer identified on the previous steps".

P8, line 175, change "assign" to "assigned", change "extracted" to "identified".

P8, lines 176-178, change this sentence to "Finally, we merged the vector polygons identified from 52 images with the attribute value of 1 in ArcGIS".

P9, line 183, change "take" to "took".

P9, line 183, as addressed in the Introduction, two studies (Li et al., 2021; Tian et al., 2013) have explored the identification of check dam on Chinese Loess Plateau. Moreover, whether the latitudes and longitudes of check dams published by Ministry of Water Resource of the People's Republic of China are available? So, the authors cannot be addressed as "there is no available spatial distribution dataset of check dams on the CLP".

Response: Thank you for your comment. These two studies mentioned were conducted within a limited area of a few hundred square kilometres, and no publicly available dataset was provided. Additionally, the government authorities only disclosed statistical data regarding the check dams without furnishing specific latitude and longitude information. Thus, currently, there is indeed no available spatial distribution dataset of check dams on the CLP.

P9, line 184, which dataset should be verified? Check dams or dam lands?

Response: We have changed this sentence to "Therefore, we took the test samples obtained from Google Earth by visual interpretation to verify the identification accuracy of dam land in our dataset." (Lines 210-211)

P9, line 185, change "traditional" to "traditionally".

P9, line 186, change "determine" to "determined". Which dataset was taken as reference? The number of check dams published by Ministry of Water Resource of the People's Republic of China?

Response: We used the check dam dataset identified in this manuscript as a reference. We have changed this sentence to "To improve the reliability of verification, we determined the number of test samples in each county according to the number of check dams in our dataset at the county level." (Lines 213-214)

P9, line 187, change "concentrated" to "dense".

P10, lines 188-189, it confuses me a lot. What did the samples be used to verify? Check dams or dam lands?

Response: We mentioned in the Method that what we extract is not the check dam (dam body), but the dam land formed by the check dam. We have replaced all the "check dam dataset" in this chapter with "dam land in our dataset".

P10, line 189, delete the second and third "the".

P10, line 190, insert "accuracy" before "(PA)".

P10, line 192, change "compare" to "compared", change "official" to "officially".

P10, line 194, change "check dam" to "dam land".

P10, line 199, there is a cited error for Zeng et al. (2022). Zeng et al. (2022a) or Zeng et al. (2022b)? delete "method".

P10, line 201, why not use the general unit of "km2"?

Response: The area of silted land of each check dam ranges from 0.01 to 625 $hm^2$ with an average of 1.8 $hm^2$, of which 50% is concentrated in 0.2–20 $hm^2$. Converting units to "$km^2$" will result in too many decimal places.

P10, line 205, change "in which" to "and", change "check dam map" to "dam land".

P10, line 206, delete "class".

P10, line 208, change "maps" to "map", change "2000" to "1947".

P11, line 210, change "combined with" to "used".

P11, line 213, delete "a slope of 1.1185 and".

P11, lines 213-215, suggesting change this sentence to "The number of check dams in this study is closer to that in official statistics reported in 2013 (58, 446) rather than that reported in 2003 (110, 000)".

P11, line 217, change "statistical" to "statistic" of the labels on the y-axis.

P11, line 218, please revise the caption of figure 6. For example, Area (a) and number (b) comparison of check dams between this study and official statistic.

P12, line 223, what does the significance of figure 7? Do the 12 sub-images represent the check dams in typical region or with special feature?

Response: We randomly selected 12 sites from the regions that exhibit dense check dam distribution on the Loess Plateau, and the identification results of dam lands at these sites are exhibited in Figure 7.

P12, line 226, change "number of" to "total", delete "are".

P12, line 229, what does "which" indicate? The spatial distribution of check dams or regional soil erosion? Suggesting split this long sentence into two short sentences.

Response: We have changed this long sentence into two short sentences: "Check dams mainly distributed in Northern Shaanxi and Western Shanxi (Fig. 7), specifically in the loess hilly and gully region. This region is known to be one of the most heavily eroded areas on the CLP and in fact, globally." (Lines 257-259)

P12, line 230, change the second "in" to "cited from", and insert the citation after "GBT16453.3-1996".

P12, line 231, delete the second "the", change "ranges from" to "of".

P12, line 232, transfer the unit of "hm2" to "km2", insert a comma after "2, 250".

P13, line 233, change "the highest" to "high".

P13, line 234, delete "number of", change "basically" to "almost", change "the results previously reported" to "the previously reported results".

P13, line 236, change "in different regions" to "at provincial level".

P13, line 237, change "Region" to "Province", change "Area" to "Silted area", change "Volume" to "Sediment volume".

P13, line 239, change "account for" to "accounting for", change "billion" to "×109".

P13, line 240, change "estimated" to "calculated", change "previously reported" to "in previous report".

P13, line 241, insert "on the CLP" before "intercept", change "billion" to "×109", change "to" to "into".

P13, line 242, delete "the period".

P13, line 243, suggesting change the x-labels with legend labels. That is to say, the x-labels are listed as Micro, Small, Medium, and Large, but legend label are listed as 0-0.2 km2, 0.2-2 km2, 2-7 km2, >7 km2.

P14, line 249, change "role" to "roles".

P14, line 250, change "of" to "among". "to date" is always located at the head or end of a sentence.

P14, line 251, change "laborious" to "laboriously", change "find check dams suitable for the research objectives" to "find suitable check dams satisfied with the research objectives".

P14, line 252, change "precise" to "precisely".

P14, line 253, change the first "of" to "for", change "provides" to "also offers".

P14, line 254, change ", which are crucial for those studies using" to "for those studies which use".

P14, lines 256-257, do not use the phrase "fill the gap" and the word "unprecedented". The authors should use euphemism in the manuscript as far as possible. Please revise this sentence.

Response: We have changed this sentence to "For policymakers, our database will provide detailed data on the spatial distribution, silted land area, and sediment volume

of check dams on the CLP." (Lines 291-292)

P14, line 260, change "billion" to "×109".

P14, line 261, delete "the period".

P14, lines 261-262, change "which equals 46% of the sediment load of the Yellow River, once the largest sediment contributor to the global ocean" to "equaling 46% of the sediment load of the Yellow River, which was once the largest sediment contributor to the global ocean".

P14, line 263, change "unprecedented" to "enormous", delete "reduction".

P14, lines 263-265, change "studies" to "article", change "emphasize" to "emphasizes", change "point" to "points", because only one citation was listed in the brackets at end of this sentence. Change "potential" to "potentiality", delete "important".

P14, line 266, change "large scale" to "macro-scale", change "or" to "with". Does literature compilation mean meta-analysis?

Response: We have changed "literature compilation" to "meta-analysis".

P14, lines 267-268, change this sentence to "Moreover, the soil silted by the check dams has higher moisture than that of terrace and slope cropland".

P14, line 268, change "shows" to "showed".

P14, line 270, change "determine" to "estimate".

P14, line 271, it is unbelievable that the Chinese government plans to build 56161 new check dams on the CLP till 2030. Because, as shown in this study, there are 50226 check dams on the CLP from 1970s to 2018. I cannot believe the government has so ambitious planning to build the nearly same check dams as that in the past 50 years during the short term of future seven years. I do not think the more check dams are good for local ecosystem. It even will break the balance between agricultural land and ecological land.

Response: Thank you for your comment. Most of the check dams on the CLP were constructed in the 1970s. Limited by the construction technology of check dams at that time, the construction efficiency of check dams was not high. According to the Outline of the Comprehensive Management Plan for the Loess Plateau (2010–2030), 56,161 new check dams will be built on the CLP in the next decade (NDRC, 2010). The construction of check dams in the gullies may not affect the ecological land because of

severe soil erosion and low vegetation coverage in the gullies. If check dams are not constructed in the gullies, only the original gully landforms will be preserved. However, in the event that check dams are constructed, the previously unavailable gullies will transform into agricultural land for planting.

Reference:

1. NDRC (National Development and Reform Commission People's Republic of China). Outline of the comprehensive management plan for the Loess Plateau (2010-2030). Beijing, China, 2010. (https://www.docin.com/p-1058786159.html, in Chinese)

P14, line 272, change "provide" to "provides".

P14, line 275, change "Combined" to "combining".

P14, line 276, delete "for the first time", change "extraction" to "identification".

P14, line 277, delete the first "the".

P14, line 278, change "extraction" to "identification".

P15, line 280, change this sentence to "and their characteristics and functions are same as small reservoirs".

P15, line 281, change "separate" to "distinguish", change "have led to" to "lead to".

P15, line 283, change this sentence to "it is difficult to obtain the images for the entire CLP in April and May of the same year".

P15, lines 285-286, change this sentence to "Therefore, we collected all available images covering most study areas in May from 2016 to 2020".

P15, lines 286-287, change the sentence in the brackets to "e.g., the check dam was constructed in 2018, but it was not imaged in 2016".

P15, line 288, change "combined" to "combing".

P15, line 289, the Sentinel-2A was launched at 23 June, 2015. Its accessible images were not earlier than that of Quickbird. Moreover, the highest spatial resolution of Sentinel-2A is 10 m, which is far lower than that of Quickbird (0.3-1.0 m). Its most advantage is for free downloading. So, if the authors want to achieve check dam dataset with higher spatial resolution, the Quickbird images are the best choice, although you

should pay much more fee on them.

Response: Thank you for your comment. Some images in Google Earth are provided by QuickBird. However, it only encompasses four conventional spectral bands (red, green, blue, and infrared), which are inadequate for machine learning-based identification of dam lands. We have changed this sentence to "In future work, our check dam dataset can be utilized as the training set, in conjunction with high-resolution satellites equipped with additional multispectral bands (e.g., Sentinel-2) and deep learning techniques, to propose an enhanced, automated, and convenient process for check dam identification." (Lines 334-339)

P15, line 290, change "extraction process" to "identification script".

P15, line 293, change the initial letter of "Digital" and "Model" to lowercase.

P15, line 301, delete "first".

P15, line 302, delete "of check dams", change "provided" to "manufactured", change "classification" to "identification", change "combined with" to "in conjunction with".

P15, line 303, change "self-developed" to "self-development".

P15, line 304, change "extraction" to "identification".

P15, line 305, change "extracted" to "identified", change "billion" to "×109"

P15, line 306, change "to" to "into", delete "the period".

P15, line 309, delete "to further verify the accuracy of this dataset".

P15, line 312, change "accuracy of extraction" to "identified accuracy", change "extraction" to "identification".

P15, line 313, change "extraction" to "identification".

P15, line 314, change "extraction" to "identification".

**Reviewer #2:**

Major concerns:

The dataset only provides silted land (dam land) data without the distribution map of check dams. Is it possible to add dam location data to this product? Otherwise, the dam lands should be used in the title and the abstract.

Response: Thank you for your comment. Besides providing data on the spatial distribution of silted land (dam land), we also present information on sediment volume and sediment retention capacity of the check dam. Moreover, once the spatial information of the dam land is available, it is very easy to locate the check dams near the dam land. Therefore, we employed the term "check dam dataset" in both the title and the abstract. We have changed the "check dam identification" in the later sections to "dam land identification".

The development of this dataset relied on two assumptions: (1) the dam lands are used for cultivation, and (2) the dam lands are distributed along river networks. Are these two assumptions enough to classify dam lands? Are there any dam lands that were not used for cultivation?

Response: Thank you for your comment. These two assumptions have been derived from comprehensive field investigations. Dam land is one of the most productive land types on the CLP, so the majority of dam land on the CLP is cultivated. The presence of a few abandoned dam lands has a limited impact on the identification of dam land. To avoid unnecessary ambiguity, we have emphasized in the method "Therefore, the identification object in this study is the cultivated dam land on the CLP, excluding water-covered dams and abandoned dam land." (Lines 132-134 in Tracked Changes, same below)

Does the dam location be considered when classifying dam lands? If yes, I think it is necessary to add dam locations to this dataset. If not, please justify how to distinguish

between dam lands and natural floodplains.

Response: Thank you for your comment. Identifying the check dam (dam body) on the CLP is a challenging task. The check dams on the CLP are constructed using various materials, such as cement, stone, and loess. Moreover, our dataset shows that the small and medium check dams with high proportion account for about 69% and 18% of the total check dams, and these small dam bodies are very difficult to identify even through visual interpretation on satellite imagery (Figure R1). Therefore, this study did not consider the dam bodies' location, but directly identified the dam lands. The dam land and natural floodplain were distinguished through the human-computer interaction program. We developed a convenient human-computer interaction program to mark non-dam land vector polygons, such as some natural floodplains and slope cropland that were captured by superimposed river networks. The left window, displaying a high-resolution Google Earth image, can be linked to the right window, which represents the potential dam lands layer identified in the previous steps. We assigned values to each identified vector polygon (e.g., non-dam land is 0, dam land is 1) based on auxiliary data, visual interpretation, and expert knowledge. Finally, we merged the vector polygons identified from 52 images with the attribute value of 1 in ArcGIS, which is the final dam land layer in the dam concentration region. We have added more detailed explanations in this section of the manuscript. (Lines 192-204)

[Figure]

Figure R1: Dam bodies of small check dams.

How did the dam lands change with time? Google images from 2016 to 2020 were used

to develop data. The dataset was compared with the official report (CMWR, 2013) to evaluate accuracy. Please provide evidence or reference to justify it is appropriate to use data in very different periods to conduct validation. I suggest clarifying the time period of this dataset. If the dam lands didn't change too much with time, how did they influence the variability of flow and sediment (Line 18)?

Response: Thank you for your comment. Figure R2 illustrates that since 2013, government departments have slowed down the construction of check dams, which led to no significant change in the number of check dams since 2013. Hence, employing government statistical data from 2013 for validation purposes is justified We have added both the manuscript and the data description in Zenodo: "This dataset is generated based on all available May images from 2016 to 2020 on the Chinese Loess Plateau at Google Earth". Despite a reduction in the number of newly constructed check dams and a decline in soil erosion rates on the Loess Plateau in recent years, the sediment retention capacity of check dams has weakened. However, when viewed from a long-term perspective, check dams have significantly influenced the flow and sediment of the Yellow River (Wang et al., 2016).

[Figure]

Figure R2: Construction time of large and medium check dams

References:

1. Liu, X., Gao, Y., Tian, Y., Li, X., and Ma, J.: Sediment Intercepted by Dams and the Sediment Production Situation Restoration of the Last 100 Years in the Yellow River Basin, Yellow River, 43, 19-23, 2021a.

2. Wang, S. A., Fu, B. J., Piao, S. L., Lu, Y. H., Ciais, P., Feng, X. M., and Wang, Y.

F.: Reduced sediment transport in the Yellow River due to anthropogenic changes, Nature Geoscience, 9, 38-41, https://doi.org/10.1038/Ngeo2602, 2016.

The slope and R used in the validation are misleading. Given the large differences among different regions (and a total number) and few data points, the slope and R are largely decided by the high values. Please consider changing the linear regression plot to a table.

Response: Thank you for your suggestion. Table 2 in the manuscript already provides similar results, and Figure may be more intuitive than Table. Consequently, in conjunction with your suggestion, we have integrated the corresponding value into Figure 6 and removed the regression analysis results (slope and $R^2$).

[Figure]

**Figure 6: Area (a) and number (b) comparison of check dams between this study and official statistics. LP: Loess Plateau; SaX: Shaanxi; SX: Shanxi; IM: Inner Mongolia; NX: Ningxia; GS: Gansu; HN: Henan; QH: Qinghai. The grey shade areas represent 95% confidence interval.**

Does Fig 3a present the dam land data developed in this study? I also see a similar figure in (Zeng et al, 2022a). Has this data already been used and published in a previous study?

Response: Figure 3a displays the dataset of dam land developed in this study. These two articles are systematic work carried out at the same time. The primary objective of the preceding publication (Zeng et al, 2022a) was to establish an empirical equation for

estimating the sediment silted by check dams. Subsequently, in this manuscript, we identified the location and the area of dam land on the Chinese Loess Plateau, and estimated the sediment volume and sediment retention capacity of the check dam using the derived empirical equation (dam land area-sediment volume). In our previous publication (Zeng et al, 2022a), we solely utilized this data for mapping the study area. Zeng, Y., Meng, X. D... Fang, N. F., and Shi, Z. H.: Estimation of the volume of sediment deposited behind check dams based on UAV remote sensing, Journal of Hydrology, 612, 128143, 2022a, https://doi.org/10.1016/j.jhydrol.2022.128143.

The authors indicated that there is a large uncertainty in the estimation of check dams according to the two reports (CMWR, 2003) and (CMWR, 2013). It seems like the two reports were released by the same institution. Is it appropriate to claim it is uncertainty just according to one very old version report and an updated version of the report? Since the result in this study is close to the new version of the report, it is weird to claim the number is still unclear.

Response: Thank you for your comment. Based on our research results, we can confirm that the recently reported data of check dams (CMWR, 2013, 58,446 check dams) is closer to the real number. But without the spatial dataset of check dams proposed in our research, it is difficult for scientific researchers and government departments to determine which report contains more reliable information on the number of check dams. Even in recent years, there are still quite a number of scientific research papers (Wang et al., 2021; Bai et al., 2023; Ran et al., 2023; Zhang et al., 2023) and government document (Yulin Municipal People's Government; 2019) using early check dam data (CMWR, 2003, 110,000 check dams), which indicates that the data in these two government reports still cause considerable uncertainty. Most importantly, the check dam data in both reports lack detailed spatial distribution information, which is the most critical factor contributing to uncertainty and the key issue to be solved in this research.

References:

1. Wang Z, Chen Z, Yu S, et al. Erosion-control mechanism of sediment check dams

on the Loess Plateau[J]. International Journal of Sediment Research, 2021, 36(5): 668-677.

2.  Bai L, Shi P, Li Z, et al. Synergistic effects of vegetation restoration and check dams on water erosion in a slope‐gully system[J]. Land Degradation & Development, 2023.

3.  Ran Q, Tang H, Wang F, et al. Numerical modelling shows an old check‐dam still attenuates flooding and sediment transport[J]. Earth Surface Processes and Landforms, 2021, 46(8): 1549-1567.

4.  Zhang J, Wang Y, Sun J, et al. Interactions between the Grain‐for‐Green Program and check dams increased vegetation carbon sequestration in the Yanhe basin, Loess Plateau[J]. Land Degradation & Development, 2023, 34(8): 2310-2321.

5.  Yulin Municipal People's Government, 2019, http://www.yl.gov.cn/xwzx/tpxw/56615.htm. (In Chinese)

Minor:

Abstract

L16-17. Wordy. Rephrase this sentence.

Response: We have rephrased this sentence to "These check dams have trapped billions of tons of eroded sediment over the past few decades, significantly reducing the sediment load of the Yellow River." (Lines 16-17)

If the number is still unclear, why the result of this study is close to the official reports? Analyzed.

Response: We have explained in the previous response. Based on our research results, we can confirm that the recently reported data of check dams (CMWR, 2013, 58,446 check dams) is closer to the real number. But without the spatial dataset of check dams proposed in our research, it is difficult for scientific researchers and government departments to determine which report contains more reliable information on the number of check dams. Even in recent years, there are still quite a number of scientific

research papers (Wang et al., 2021; Bai et al., 2023; Ran et al., 2023; Zhang et al., 2023) and government document (Yulin Municipal People's Government; 2019) using early check dam data (CMWR, 2003, 110,000 check dams), which indicates that the data in these two government reports still cause considerable uncertainty. Most importantly, the check dam data in both reports lack detailed spatial distribution information, which is the most critical factor contributing to uncertainty and the key issue to be solved in this research.

R-value is misleading.

Response: We have removed the R-value.

Introduction

Do you mean the whole country is a study area?

Response: Our study area is the Chinese Loess Plateau. However, check dams are exclusively distributed on the Chinese Loess Plateau, and do not exist in other regions of China. Therefore, it can also be regarded that we study the check dams of the whole China.

Variations

Response: We have changed "variation" to "variations"

L74-75. Revise this sentence. Maybe "Currently, only two studies…". Delete the repeated "check dams".

Response: We have rephrased this sentence to "However, currently only a few studies have explored the possibility of obtaining check dam data based on remote sensing technology". (Lines 78-79)

Methods

Please use past tense when describing the methods. Rephrase this sentence.

Response: We have performed detailed proofreading, checking spelling, grammar,

sentence structure, and terminology through native English experts in this field.

L195-200. Such a simple and empirical method was used to estimate sediment volume. Please discuss the uncertainty of this method. Can this method represent large-scale conditions? What about other impact factors, such as slope?

Response: Thank you for your comment. In our previous study published in the Journal of Hydrology (Zeng et al., 2022), we have discussed in detail the method and accuracy of sediment volume estimation of check dams. We combine unmanned aerial vehicle (UAV) photogrammetry and simulate submerging analysis to propose an empirical formula for check dam volume estimation. We obtained 1339 groups of different topographic factors (Figure R3) and ascertained the corresponding volumes of sediment deposited behind check dams, and established five different models combined with regression analysis (Figure R4). Two different sets of data were used for method validation and optimal model determination. The results showed that the error of the optimal model in the volume estimation of single check dam and regional check dams is 12–13% and 2–3%, respectively. Additionally, the area-volume model has the potential to evaluate the sediment retention capacity of check dams (in the order of billions of cubic meters) in the whole LHRC, because the variables are easy to obtain and the model accuracy is relatively high. More details about this method can be found in Zeng et al. (2022).

[Figure]

Figure R3. Correlation analysis among the sediment volume and topographic factors.
Circle colour and size correspond to the strength of the correlation

[Figure]

Figure R4. Comparison between different volume estimation methods. The volume of the X-axis represents the real volume obtained by high-resolution DEM or previous literature. The estimated volume of the Y-axis is the predicted sediment volume obtained by different fitting equations.

Reference:

1. Zeng, Y., Mang, X.D., Zhang, Y., Dai, W., Fang, N.F., Shi, Z.H., 2022. Estimation of the volume of sediment deposited behind check dams based on UAV remote sensing. Journal of Hydrology, 612: 128143. DOI:10.1016/j.jhydrol.2022.128143

Results and discussion

"Compared"? "We provide the check dam dataset on the CLP for the first time by combining high-resolution and easily accessible Google Earth images and object-based classification strategy".

Response: Thank you for your suggestion. Based on your and another reviewer's suggestions, we have revised this sentence to summarize the advantages of our research. "This study provides the spatial distribution dataset of check dams at the regional scale

on the CLP for the first time. The object-based classification method used in this study demonstrated high accuracy and good visual effect, especially in combination with the self-development computer program. Moreover, besides providing spatial distribution information, we also offer data on sediment volume and sediment retention capacity of the check dam. This information is crucial, yet currently lacking, and it can serve as valuable references for regional soil and water conservation planning." (Lines 313-319)

**Reviewer #3:**

The introduction needs improvement in terms of logical flow and structure. It should briefly introduce soil erosion as a significant environmental issue and its threat to sustainable development. Transition to the measures addressing soil erosion, including dam construction. Highlight the advantages of dam construction in arid regions, such as soil retention and erosion prevention. Mention additional benefits like carbon sequestration and food supply. Discuss the lack of accurate data on dam numbers and spatial distribution, posing challenges in assessing their impact on sediment transport. Finally, clarify the study's objectives, methods, and innovation, emphasizing the creation of a vectorized dataset using high-resolution imagery for assessing ecosystem services and informing conservation projects. A more coherent and concise introduction would improve readability and convey the research's significance.

Response: Thank you for your suggestion. We have revised our introduction according to your suggestion and the following logic:

**In paragraph 1,** we first introduced the *possible environmental impacts of soil erosion,* and t then discussed *some measures to mitigate soil erosion,* thus drawing forth the study object (check dams). Then, we *emphasized the benefits of check dams* in sediment retention, carbon sequestration, and grain supply.

**In paragraph 2,** we first *introduced our study area,* the Chinese Loess Plateau (CLP), which is also the most densely distributed area of check dams in the world. We *highlighted the absence of essential data* regarding the numbers and spatial distribution of check dams on the CLP, *emphasizing the significance and necessity of a comprehensive check dam dataset.*

**In paragraph 3,** we *reviewed the current research status and shortcomings* of check dam identification.

**In paragraph 4,** we first *discussed the reasons why there is no available dataset of check dams on the CLP* (or the current research difficulties). Then we *emphasized the advantages and applicability* of the object-based classification method. Finally, we *summarized our research objectives, methods, and innovative significance.*

The section on data and methodology in the paper is comprehensive. However, it is essential to provide a detailed description of the data collection process, including data sources, collection methods, and tools used. For example, relevant details like the timing of image acquisition should be included. Additionally, the overall study design and methodology should be explained, highlighting the chosen research methods and the reasons behind their selection. For instance, it is important to clarify why an object-oriented approach was adopted for image segmentation and discuss the model parameters. In conclusion, emphasizing the effectiveness and advantages of the employed data and methods is crucial.

Response: Thank you for your suggestion. We have made detailed modifications to the data and methodology according to your suggestions. We mentioned that we have collected all available images from May 2016 to 2020 in the dam dense region in Google Earth, with spatial resolution of 0.3-1.0 m. A significant change in the dam land area can occur when the time span is extensive, resulting in a higher margin of error in identifying the dam land. This work started around 2018, so we chose the images from 2016 to 2020.

*We have explained in the introduction the reasons why we chose the object-based classification method.* "Firstly, check dams on the CLP are mainly small and medium-sized check dams, with a correspondingly silted area of 0.2-2 $hm^2$. The decametric-resolution images (e.g., Landsat-7/8) may generate erroneous judgment for identification of silted land due to the limited number of pixels and jagged edges. Secondly, the silted land formed by check dams is usually used for planting crops, which is difficult to be distinguished from the surrounding slope cropland and terrace in terms of spectral characteristics. Finally, silted land has the characteristics of large spatial heterogeneity and patch fragmentation, which is more difficult to be identified than other land use types. The traditional pixel-based identified methods usually focus on medium resolution images, seldom refer to the structure and texture characteristics and the correlative information between adjacent pixels. Meanwhile, these methods also emerge salt-and-pepper noise, which reduces the accuracy of classification. In contrast, the object-based classification method comprehensively considers a series of

factors, such as spectral features, shape, size, texture, and adjacency, can obtain high-precision identified results combing with high spatial resolution images. Therefore, we attempt to identify the check dams on the CLP using object-based classification method in conjunction with high spatial resolution (0.3-1.0 m) and easily accessible Google Earth images in this study. The self-development computer program in conjunction with auxiliary data, visual interpretation, and expert knowledge are used to improve the identified accuracy of check dams." (Lines 86-105 in Tracked Changes, same below)

*The model parameters include segmentation parameters and classification parameters.* "The scale, shape, and compactness are the most important parameters in multiresolution segmentation algorithms, and their values will affect the segmentation results (Munyati, 2018). The scale parameter is used to determine the maximum heterogeneity of the generated object and to control the size of the segmented object. The estimation of scale parameter 2 (ESP2) plugin in eCognition Developer can automatically evaluate the segmentation effect based on the local variance (LV) and its rate of change (ROC). Therefore, we used the ESP2 plugin to determine the optimal scale parameter for dam land identification. The setting of shape and compactness is also crucial for the segmentation and subsequent identification of dam land, as the dam land is usually narrow and irregular in shape. Therefore, we made different combinations of these parameters to test the best parameter settings in combination with visual inspection. Finally, we set the scale parameter, shape weight, and compactness weight to 100, 0.7, and 0.3 to obtain the best segmentation results." (Lines 162-172)

*We used the assigned class algorithm in the eCognition Developer to determine the classification parameters.* "When there were significant differences between the target and background categories for some features, the assigned class algorithm could be used to construct identification rules. The assign class algorithm was more suitable for our study than other identification methods, as our goal is to identify more dam land in a target category. For each segmented image, we first randomly selected 30-80 dam lands and non-dam lands at different locations according to the amplitude of the image. Then, we used the separability and thresholds algorithm in eCognition Developer to automatically select identification features (e.g., red, green, and blue band, shape,

texture, and brightness) and determine the threshold of selected features. Finally, we used the feature threshold to classify the segmented objects and manually adjusted the threshold range using visual interpretation to ensure that all dam lands were included in the threshold range. Through the above steps, we obtained the bare land layer in the dam dense region in May, mainly including the dam land in gullies and the cropland on slope land." (Lines 179-190)

The results and discussion section of the paper should be approached with attention to the following aspects. It is important to integrate the results with the discussion, providing interpretations of the findings and presenting evidence that supports or contradicts the research hypotheses. The main discoveries of the study should be summarized, and their significance and contribution to the relevant field should be assessed and discussed. Furthermore, I would suggest the author to include a discussion on the strengths of the present study.

Response: Thank you for your suggestion. We have made detailed modifications to the results and discussion according to your suggestions. The dataset we proposed is the first check dam spatial distribution dataset on the Chinese Loess Plateau, and there is currently no dataset available for comparison. Therefore, we focus on the validation of the accuracy of the dataset and the presentation of the main findings of the dataset. We highlight our research significance and contribution in Section 3.3 Broader implications. (Lines 281-311)

1. Our dataset can provide the precisely spatial location for check dams, which greatly *saves the manpower and material resources of researchers.* More importantly, our dataset also *offers key parameters* such as silted land area and sediment volume of check dams for those studies which *use check dams to calculate soil erosion rate, estimate watershed sediment delivery ratio, and validate the soil erosion model*.

2. For policymakers, our database will provide *detailed data on the spatial distribution, silted land area, and sediment volume of check dams* on the CLP. When these key parameters of check dams are obtained, we can conveniently and *accurately evaluate the ecosystem services functions of check dams,* including sediment

retention, carbon sequestration, and grain supply.

3. Finally, according to the Chinese government's plan, 56,161 new check dams will be built on the CLP by 2030. Our spatial distribution data of check dams *provides important information for optimizing the location of the new check dams* in the future.

We have summarized and added our strengths in Section 3.4 (Lines 313-319):

1. This study provides the spatial distribution dataset of check dams *at the regional scale on the CLP for the first time*. As far as we know, there is no available dataset of spatial distribution of check dams, either from researchers or government departments.

2. Although we used a semi-automatic method, this method *has high accuracy and good visual effect.* When combined with our self-developed human-computer interaction software, the identification accuracy of check dams can be significantly improved. We believe that *ensuring accuracy is even more important* for the basic dataset provided for the first time, so we have added more visual interpretation and expert experience in the steps of identification feature selection and human-computer interaction.

3. Besides providing *spatial distribution information*, we also offer data on *sediment volume and sediment retention capacity* of the check dam. This information is *crucial, yet currently lacking*, and it can serve as valuable reference for regional soil and water conservation planning.